# Hepatic *MIR20B* promotes nonalcoholic fatty liver disease by suppressing *PPARA*

**Yo Han Lee[1†], Hyun-Jun Jang[1†], Sounkou Kim[1†], Sun Sil Choi[1], Keon Woo Khim[1], Hye-jin Eom[1], Jimin Hyun[1], Kyeong Jin Shin[1], Young Chan Chae[1], Hongtae Kim[1], Jiyoung Park[1], Neung Hwa Park[2], Chang-Yun Woo[3], Chung Hwan Hong[4], Eun Hee Koh[3], Dougu Nam[1]\*, Jang Hyun Choi[1]\***

[1]Department of Biological Sciences, Ulsan National Institute of Science and Technology, Ulsan, Republic of Korea; [2]Department of Internal Medicine, Ulsan University Hospital, Ulsan, Republic of Korea; [3]Department of Internal Medicine, Asan Medical Center, Seoul, Republic of Korea; [4]Department of Medical Science, Asan Medical Center, Seoul, Republic of Korea

**\*For correspondence:**
dougnam@unist.ac.kr (DN);
janghchoi@unist.ac.kr (JHyunC)

[†]These authors contributed equally to this work

**Competing interest:** The authors declare that no competing interests exist.

## Abstract

**Background:** Non-alcoholic fatty liver disease (NAFLD) is characterized by excessive lipid accumulation and imbalances in lipid metabolism in the liver. Although nuclear receptors (NRs) play a crucial role in hepatic lipid metabolism, the underlying mechanisms of NR regulation in NAFLD remain largely unclear.

**Methods:** Using network analysis and RNA-seq to determine the correlation between NRs and microRNA in human NAFLD patients, we revealed that *MIR20B* specifically targets *PPARA. MIR20B* mimic and anti-*MIR20B* were administered to human HepG2 and Huh-7 cells and mouse primary hepatocytes as well as high-fat diet (HFD)- or methionine-deficient diet (MCD)-fed mice to verify the specific function of *MIR20B* in NAFLD. We tested the inhibition of the therapeutic effect of a PPARα agonist, fenofibrate, by *Mir20b* and the synergic effect of combination of fenofibrate with anti-*Mir20b* in NAFLD mouse model.

**Results:** We revealed that *MIR20B* specifically targets *PPARA* through miRNA regulatory network analysis of nuclear receptor genes in NAFLD. The expression of *MIR20B* was upregulated in free fatty acid (FA)-treated hepatocytes and the livers of both obesity-induced mice and NAFLD patients. Overexpression of *MIR20B* significantly increased hepatic lipid accumulation and triglyceride levels. Furthermore, *MIR20B* significantly reduced FA oxidation and mitochondrial biogenesis by targeting *PPARA*. In *Mir20b*-introduced mice, the effect of fenofibrate to ameliorate hepatic steatosis was significantly suppressed. Finally, inhibition of *Mir20b* significantly increased FA oxidation and uptake, resulting in improved insulin sensitivity and a decrease in NAFLD progression. Moreover, combination of fenofibrate and anti-*Mir20b* exhibited the synergic effect on improvement of NAFLD in MCD-fed mice.

**Conclusions:** Taken together, our results demonstrate that the novel *MIR20B* targets *PPARA*, plays a significant role in hepatic lipid metabolism, and present an opportunity for the development of novel therapeutics for NAFLD.

**Funding:** This research was funded by Korea Mouse Phenotyping Project (2016M3A9D5A01952411), the National Research Foundation of Korea (NRF) grant funded by the Korea government (2020R1F1A1061267, 2018R1A5A1024340, NRF-2021R1I1A2041463, 2020R1I1A1A01074940, 2016M3C9A394589324), and the Future-leading Project Research Fund (1.210034.01) of UNIST.

## Editor's evaluation

The manuscript by Lee et al. provides mechanistic insight into the regulatory role of micro RNAs in modulating nuclear receptor expression and function. This is likely to have a high impact on the field as nuclear receptor regulation of metabolic disease is well established, however, the molecular mechanisms governing this process still remains unknown largely. Lee et al.'s manuscript provides a molecular target (miR-20b) that holds therapeutic potential in improving hepatic steatosis.

## Introduction

Obesity has emerged as a host of metabolic disorders, such as non-alcoholic fatty liver disease (NAFLD). Many reports have demonstrated that 90% of obese patients in the United States have NAFLD ranging from hepatic steatosis to much more severe forms of non-alcoholic steatohepatitis (NASH), which can induce fibrosis, cirrhosis, and hepatocellular carcinoma (HCC) (*Corey and Kaplan, 2014*). NAFLD is associated with hepatic metabolic reprogramming that leads to excessive lipid accumulation and imbalances in lipid metabolism in the liver (*de Alwis and Day, 2008*). Hepatic lipid homeostasis is appropriately described as a complex machinery involving signaling and transcriptional pathways, as well as targeted genes associated with fatty acid (FA) oxidation and lipogenesis (*Fabbrini et al., 2010*). Although the pathogenesis of NAFLD has been widely studied for years, the molecular mechanisms underlying its complicated disorder are still being investigated.

Nuclear receptors (NRs) are a superfamily of ligand-activated transcription factors that regulate biological homeostasis (*McKenna et al., 2009*). Recent evidence suggests that NRs are key regulators in the progression of diverse hepatic diseases associated with glucose and lipid metabolism, inflammation, bile acid homeostasis, fibrosis, and cancer development in the liver (*López-Velázquez et al., 2012*). Among them, growing evidence suggests a link between PPARα and obesity-induced NAFLD. Hepatic PPARα plays an important role in energy homeostasis by regulating the expression of genes required for FA uptake, FA oxidation, and triglyceride (TG) hydrolysis in the liver (*Chakravarthy et al., 2005*). The decreased expression of PPARα is significantly associated with severity in NAFLD patients (*Francque et al., 2015*). Therefore, understanding the molecular mechanism underlying PPARα regulation is critical for understanding the pathogenesis of NAFLD.

MicroRNAs (miRNAs) are short, non-coding RNA molecules with a length of 18–25 nucleotides that play an important role in regulating the expression of target genes in a post-transcriptional manner by targeting base-pairing with the 3'UTR of specific target mRNAs, inhibiting translation, or mRNA degradation (*Bartel, 2004*). These miRNAs contribute to the regulation of a wide variety of cellular functions and metabolic homeostasis, including fatty acid metabolism. Recent studies have suggested that miRNAs significantly regulate the pathogenesis of NAFLD by targeting the nuclear receptors (*López-Sánchez et al., 2021*). Previous report has demonstrated that *MIR20B*, a member of the MIR17 family, presents in the circulating plasma of NAFLD patients and has been highlighted as a novel biomarker of NAFLD and type 2 diabetes mellitus (T2DM) for the diagnosis and risk estimation of NAFLD (*Ye et al., 2018*). However, the mechanisms underlying the involvement of *MIR20B* in the occurrence and progression of NAFLD remain unknown. In this study, we analyzed the regulatory networks of miRNAs for NR genes and RNA-seq data in NAFLD patients, which prioritized *MIR20B* as a key regulator in NAFLD.

## Materials and methods

**Key resources table**

| Reagent type (species) or resource | Designation | Source or reference | Identifiers | Additional information |
|---|---|---|---|---|
| Genetic reagent (*Mus. Musculus*) | C57BL/6JbomTac | DBL | RRID:IMSR_TAC:b6jbom | |
| Cell line (*Homo sapiens*) | HepG2 | ATCC | HB-8065, RRID:CVCL_0027 | |

*Continued on next page*

*Continued*

| Reagent type (species) or resource | Designation | Source or reference | Identifiers | Additional information |
|---|---|---|---|---|
| Cell line (*Homo sapiens*) | Huh-7 | Dr. Yoshiharu Matsuura; originally from Japanese Collection of Research Bioresources Cell Bank | JCRB0403, RRID:CVCL_0336 | |
| Antibody | Anti-PPARα (Rabbit Polyclonal) | abcam | Cat# ab24509, RRID:AB_448110 | WB (1:1000) |
| Antibody | Anti-HSP90 (Rabbit Polyclonal) | Cell Signaling Technology | Cat# 4,877 S, RRID:AB_2233307 | WB (1:1000) |
| Sequence-based reagent | *MIR20B*/*Mir20b* (*MIR20B*/*Mir20b* mimic) | GenePharma | N/A | Sequence: CAAAGUGCUCAUAGUGCAGGUAG |
| Sequence-based reagent | anti-*MIR20B*/*Mir20b* (*MIR20B*/*Mir20b* inhibitor) | GenePharma | N/A | Sequence: CUACCUGCACUAUGAGCACUUUG |
| Sequence-based reagent | *PPARA* siRNA | GenePharma | N/A | Sequence: CGGCGAGGATAGTTCTGGAAGCTTT |
| Sequence-based reagent | *PPARA* shRNA | Sigma-Aldrich | N/A | Sequence: GAACAGAAACAAATGCCAGTA Sequence: GTAGCGTATGGAAATGGGTTT |
| Sequence-based reagent | Primers for qPCR | This paper | N/A | |
| Recombinant DNA reagent | psiCHECK-2- *PPARA*-WT (plasmid) | This paper | N/A | |
| Recombinant DNA reagent | psiCHECK-2- *PPARA*-Mut (plasmid) | This paper | N/A | |
| Recombinant DNA reagent | pOTTC385-pAAV CMV-IE IRES EGFP-*Mir20b* | This paper | N/A | |
| Recombinant DNA reagent | pOTTC385-pAAV CMV-IE IRES EGFP-anti-*Mir20b* | This paper | N/A | |
| Commercial assay or kit | Dual-Luciferase kit | Promega | Cat# E1910 | |
| Commercial assay or kit | RNeasy mini kit | Qiazen | Cat# 74,004 | |
| Commercial assay or kit | QuickChange II Site-Directed Mutagenesis Kit | Agilent | Cat# 200,521 | |
| Commercial assay or kit | AAVpro Purification Kit | Takara Bio. | Cat# 6,675 | |
| Commercial assay or kit | Triglyceride Colorimetric Assay Kit | Cayman Chemical | Cat# 10010303 | |
| Commercial assay or kit | Alanine Transaminase Colorimetric Activity Assay Kit | Cayman Chemical | Cat# 700,260 | |
| Commercial assay or kit | Aspartate Aminotransferase Colorimetric Activity Assay Kit | Cayman Chemical | Cat# 701,640 | |
| Commercial assay or kit | Mouse Insulin ELISA Kit | Crystal Chem | Cat# 90,080 | |
| Commercial assay or kit | PicoSens Free Fatty Acid Quantification Kit | BioMax | Cat# BM-FFA-100 | |

*Continued on next page*

Continued

| Reagent type (species) or resource | Designation | Source or reference | Identifiers | Additional information |
|---|---|---|---|---|
| Commercial assay or kit | β-hydroxybutyrate assay Kit | abcam | Cat# ab83390 | |
| Chemical Compound, drug | Oleic acid | Sigma-Aldrich | Cat# O1008 | |
| Chemical Compound, drug | Fenofibrate | Santa Cruz biotechnology | Cat# sc-204751 | • HepG2 cells were treated with fenofibrate (100 μM)<br>• mice injected with fenofibrate (100 mg/kg) |

## Cell culture

Human liver cells, HepG2 and Huh-7 cells were purchased from American Type Culture Collection (ATCC, Manassas, VA) and cultured in Dulbecco's Modified Eagle's Medium (DMEM) containing 10% fetal bovine serum (Gibco, BRL, Grand Island, NY) and 1% penicillin/streptomycin (ThermoFisher Scientific, Waltham, MA). miRNAs and siRNA were obtained from GenePharma (Shanghai, China). The miRNAs and siRNA used in this study are listed in the table below. HepG2 and Huh-7 cells were transfected with miRNA or siRNA using Lipofectamine RNAiMAX transfection reagent (ThermoFisher Scientific) according to the manufacturer's instructions and following experiments were performed 48 hr after transfection. For intracellular lipid accumulation, cells were cultured in a medium with the addition of 1 mmol/L sodium oleic acid for 24 hr and then cells were harvested for further analysis. Images of cells stained with Oil Red O were obtained with EVOS FL (Thermo Fisher Scientific). For quantification, Oil Red O was extracted with isopropanol and absorbance was measured at 520 nm (*Lim et al., 2012*).

| Name | Sequence |
|---|---|
| MIR NC | UCACAACCUCCUAGAAAGAGUAGA |
| *MIR20B*/*Mir20b* (Mimic) | CAAAGUGCUCAUAGUGCAGGUAG |
| anti-*MIR20B*/*Mir20b* (Inhibitor) | CUACCUGCACUAUGAGCACUUUG |
| *siPPARA* | CGGCGAGGATAGTTCTGGAAGCTTT |

## Human patients

Human liver tissue samples of 21 patients were acquired from the BioResource Center (BRC) of Asan Medical Center, Seoul, Republic of Korea. The process of 21 human tissue samples was officially approved by the Institutional Review Board of Asan Medical Center (IRB approval number: 2018–1512). Human samples used in this study were from hepatocellular carcinomas (HCC) liver resection. All samples were taken from non-tumor parts of the resected tissue and selected as 11 normal liver samples, 4 simple steatosis, and 6 NASH samples through histological analysis of pathologists using the grading and staging system for NASH (*Brunt et al., 1999*). Although limitations in completely excluding the effect of HCC, histologically normal tissue adjacent to the tumor has obvious distinctions in global transcriptome patterns for the tumor and can be used as a control (*Aran et al., 2017*). There were no significant differences in age and male/female ratio among the three groups. Specifically, serum AST, ALT, and fasting glucose levels were significantly increased with progression from normal to NASH, but total cholesterol was comparable as previously reported (*Chung et al., 2020*). In addition, all patients diagnosed with alcoholic liver disease, viral infected hepatitis, and toxic hepatitis were excluded.

## Library preparation for transcriptome sequencing

RNA-seq was performed on triplicate sample from HepG2 cell with or without overexpression of *MIR20B*. Total RNA was isolated using the RNeasy mini kit (Qiagen, Hilden, Germany) according to the manufacturer's instructions. Library prep and RNA-seq were performed by Novogene (Hong Kong). A total amount of 1 μg RNA per sample was used as input material for the RNA sample preparations.

Sequencing libraries were generated using NEBNext UltraTM RNA Library Prep Kit for Illumina (NEB, Ipswich, MA) following manufacturer's recommendations and index codes were added to attribute sequences to each sample. PCR products were purified (AMPure XP system, Beckman Coulter Life Sciences, Indianapolis, IN) and library quality was assessed on the Agilent Bioanalyzer 2,100 system (Agilent Technologies, Inc, Santa Clara, CA). The clustering of the index-coded samples was performed on a cBot Cluster Generation System using PE Cluster Kit cBot-HS (Illumina, San Diego, CA) according to the manufacturer's instructions. After cluster generation, the library preparations were sequenced on an Illumina platform (NovaSeq 6000 PE150) and paired-end reads were generated.

## miRNA regulatory network analysis of nuclear receptor genes

The RNA-seq fold change data for 16,010 genes were obtained from the Supplementary file 1 of a previous work (*Hoang et al., 2019*). The miRNA target genes that shared evolutionarily conserved binding sites for 353 miRNAs ($\geq 10$ *genes* were downloaded from TargetScan database (Ver.7.2)) (*Agarwal et al., 2015*). We used the 50 genes that belonged to the nuclear receptor transcription pathway in REACTOME database (*Jassal et al., 2020*; *Liberzon et al., 2015*) to analyze how miRNAs regulate the transcription of NR genes. Among them 17 genes were significantly downregulated in NASH patients (adjusted p-value < 0.1). The enrichment of these downregulated NR genes in the targets of the 353 miRNAs were assessed using hypergeometric distribution. The p-value of a miRNA indicates the probability of having downregulated nuclear receptors among the miRNA targets. The p-value is given as follows:

$$\text{p-value}\left(miRNA\right) = \sum_{i=O}^{M \wedge D} \frac{\binom{D}{i}\binom{N-D}{M-i}}{\binom{N}{M}}$$

where *N* is the total number of genes analyzed, *M* is the number of candidate target genes of the miRNA, *D* is the downregulated NR genes, and *O* is the observed overlap between miRNA targets and the downregulated NR genes.

The miRNAs whose target genes were enriched in the downregulated NR genes with adjusted p-value ≤ 0.05 were used to construct the regulatory networks of the NR transcription pathway (*Figure 1A*).

## Differential expression and gene-set enrichment analysis (GSEA)

The differential expression analysis of RNA-seq data were performed using limma package (*Ritchie et al., 2015*) where moderated *t*-test was applied for voom-transformed read counts. The resulting fold-change values between the test and control conditions were used for the pathway analysis. The preranked GSEA (R package) (*Subramanian et al., 2005*) was used for the pathway analysis of gene sets from WikiPathway (*Martens et al., 2021*), REACTOME (*Jassal et al., 2020*), KEGG (*Kanehisa et al., 2017*) databases (MSigDB) and the enrichment score plots.

## Immunoblotting

Supernatants containing protein contents were determined by Bradford Protein Assay (Bio-Rad Laboratories, Hercules, CA). Proteins were immunoblotted with anti-PPARα (ab24509, Abcam, Cambridge, MA) and anti-HSP90 (4,877 S, Cell Signaling Technology, Danvers, MA).

## Quantitative PCR

Total mRNAs were isolated using TRIzol reagent purchased from Thermo Fisher Scientific. Reverse-transcription of the RNA was performed with ABI Reverse Transcription Kit (Thermo Fisher Scientific). Quantitative PCR was performed using 7900HT Fast Real-Time PCR System (Life Technologies, Carlsbad, CA) following the manufacturer's instructions. Relative mRNA expression levels of each gene were normalized to TATA-binding protein TBP. The mtDNA copy number was evaluated based on the ratio of mtDNA to nuclear DNA by quantitative PCR. The mtDNA was quantified based on the

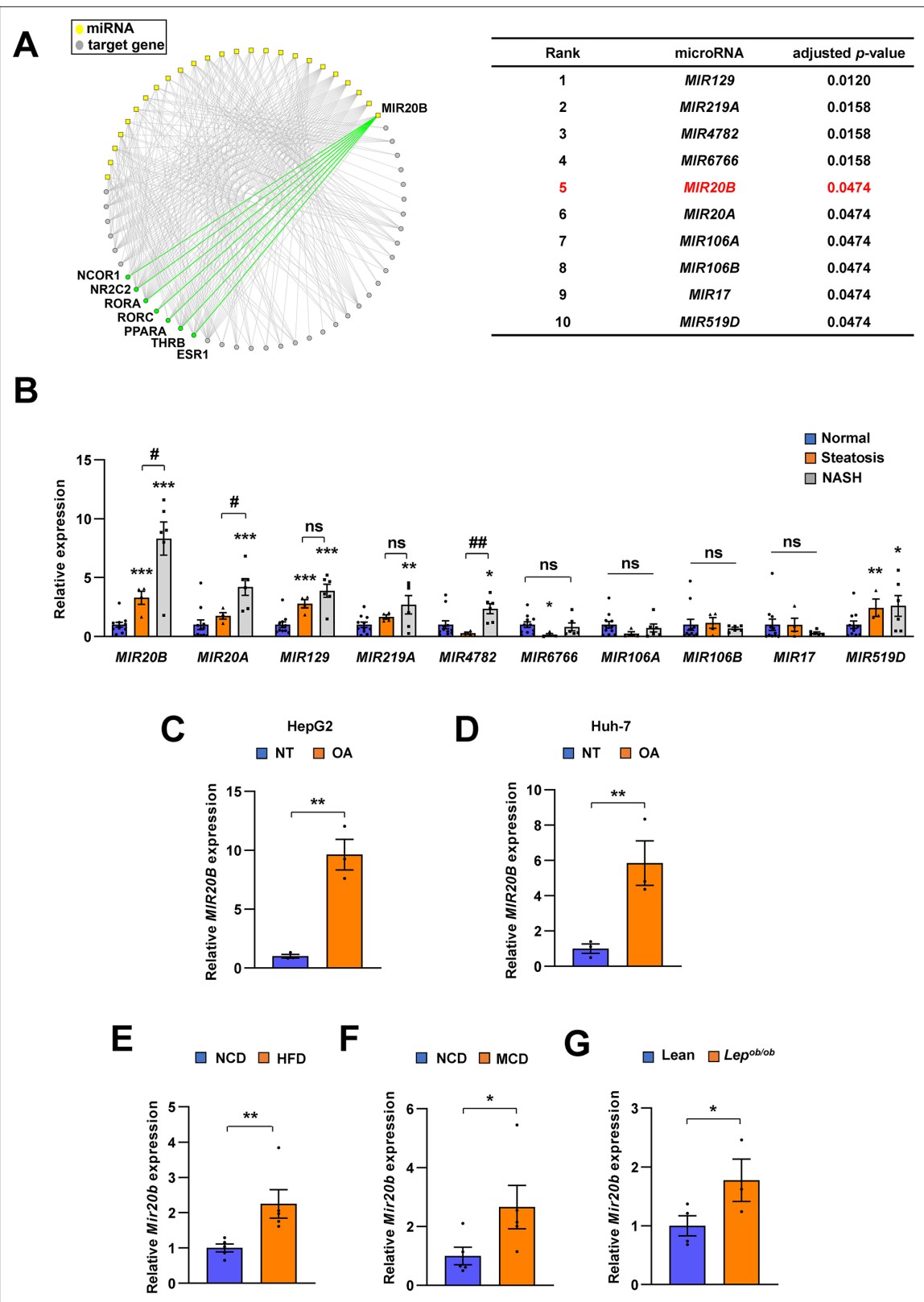

**Figure 1.** *MIR20B* expression is significantly increased in the livers of dietary obese mice and humans. The miRNA regulatory networks for NR genes downregulated in the transcriptome of NAFLD patients. The adjusted p-values in the table represent the enrichment of miRNA targets in the downregulated NR genes (hypergeometric distribution) (**A**). The expression of miRNAs was measured in indicated condition by quantitative RT-PCR (**B–G**). Hepatic miRNA levels of patients with steatosis (n = 4) or NASH (n = 6) were normalized to those of normal individuals (n = 11). *p < 0.05, **p

*Figure 1 continued on next page*

*Figure 1 continued*

< 0.01 and \*\*\*p < 0.001 vs normal individuals. #p < 0.05 and ##p < 0.01 vs patients with steatosis (**B**). *MIR20B* levels from HepG2 cells (**C**) and Huh-7 cells (**D**) treated with OA for 24 hr were normalized to no treatment (NT). Hepatic *Mir20b* levels from C57BL/6 J mice fed high-fat diet (HFD) (**E**), or methionine-deficient diet (MCD) (**F**) were normalized to normal chow diet (NCD). Hepatic *Mir20b* levels from leptin-deficient *Lep^{ob/ob}* mice were normalized to lean wild mice (**G**). Values represent means ± SEM (n = 3–5). \*p < 0.05, \*\*p < 0.01 vs NT in cells or NCD-fed mice. Raw data can be found in a Source Data file named '*Figure 1—source data 1*'.

The online version of this article includes the following source data and figure supplement(s) for figure 1:

**Source data 1.** *MIR20B* expression is significantly increased in the livers of dietary obese mice and humans.

**Figure supplement 1.** Clinical characteristics of control individuals (N = 11), steatosis (steatosis >50%, N = 4) and NASH patients (N = 6).

**Figure supplement 2.** Pearson correlation between clinical characteristics of patients and *MIR20B* expression.

**Figure supplement 3.** Hepatic gene expression associated with steatosis, inflammation and fibrosis is increased in dietary or genetic obese mice.

**Figure supplement 3—source data 1.** Hepatic gene expression associated with steatosis, inflammation and fibrosis is increased in dietary or genetic obese mice.

**Figure supplement 4.** *MIR20B* expression significantly increases in OA-treated hepatocytes and the livers of dietary obese mice.

**Figure supplement 4—source data 1.** MIR20B expression significantly increases in OA-treated hepatocytes and the livers of dietary obese mice.

mitochondrial gene, *VIPR1*, and *MT-ATP6*, respectively. The relative amounts of mtDNA were normalized to nuclear DNA, *B2M*. The primer pairs used in this study are listed in the table below.

| | Sequences of the primers for qPCR | |
|---|---|---|
| Human Gene | Forward primer | Reverse primer |
| *MIR20B* | GCAAAGTGCTCATAGTGCAGGTAG | TCGCACTTGTCATACACCAG |
| *MIR129* | CTTTTTGCGGTCTGGGCTTGC | AGTGCAGGGTCCGAGGTATT |
| *MIR219A* | TGATTGTCCAAACGCAATTC | GAACATGTCTGCGTATCTC |
| *MIR4782* | TTCTGGATATGAAGACAATCAA | CAAGATCTATACTTCTGTTAGT |
| *MIR6766* | CGGGTGGGAGCAGATCTTAT | GAACATGTCTGCGTATCTC |
| *MIR20A* | TGCGCTAAAGTGCTTATAGTGC | CCAGTGCAGGGTCCGAGGTATT |
| *MIR106A* | GATGCTCAAAAAGTGCTTACAGTGCA | TATGGTTGTTCTGCTCTCTGTCTC |
| *MIR106B* | CAAGTACCCACAGTGCGGT | CTCGCTTCGGCAGCACA |
| *MIR17* | TCTAGATCCCGAGGACTG | ATCGTGACCTGAACC |
| *MIR519D* | AGTGCCTCCCTTTAGAG | GAACATGTCTGCGTATCTC |
| *U6* | CTCGCTTCGGCAGCACA | AACGCTTCACGAATTTGCGT |
| *PPARA* | GCTATCATTACGGAGTCCACG | TCGCACTTGTCATACACCAG |
| *CPT1A* | AGGCGACATCAATCCGAAC | AAAGGCTACGAATGGGAAGG |
| *ACOX1* | CCACGTATGACCCTGAAACC | TCCATAGCATTTCCCCTTAGTG |
| *SREBF1* | CAACACAGCAACCAGAAACTC | CTCCACCTCAGTCTTCACG |
| *FASN* | CAAGCTGAAGGACCTGTCTAG | CGGAGTGAATCTGGGTTGATG |
| *CD36* | GCCAGGTATTGCAGTTCTTTTC | TGTCTGGGTTTTCAACTGGAG |
| *FABP1* | GCAGAGCCAGGAAAACTTTG | AGCGGTGATGGTGAACTTG |
| *PPARGC1A* | ACCAAACCCACAGAGAACAG | GGGTCAGAGGAAGAGATAAAGTTG |
| *SIRT1* | CCCTCAAAGTAAGACCAGTAGC | CACAGTCTCCAAGAAGCTCTAC |
| *VIPR1* | CTCCACCATTAGCACCCAAAGCTAAG | GATATTGATTTCACGGAGGATGGTGGTC |
| *MT-ATP6* | AACGAAAATCTGTTCGCTTCAT | ATGTGTTGTCGTGCAGGTAGAG |
| *HMGCL* | GAGTTTTCAGAGGTTTGACGC | CAAGAGCACAGGAGACGTAC |

*Continued on next page*

*Continued*

| | Sequences of the primers for qPCR | |
|---|---|---|
| ACAT1 | CGGGCTAACTGATGTCTACAA | CAAATTTCCCAGCTTCCCATG |
| ACAT2 | CCCAGAACAGGACAGAGAATG | AGCTTGGACATGGCTTCTATG |
| RORA | GGTGATGCTTTTGTTCTTACTGG | TGTCTCCACAGATCTTGCATG |
| RORC | TGGTGCTGGTTAGGATGTG | GGAGTGGGAGAAGTCAAAGATG |
| THRB | CATCAAAACTGTCACCGAAGC | TCCAAGTCAACCTTTCCACC |
| NRBP1 | GTTCCACCCAGCATTGTTTG | CAGGGATTTCAGCCAGTACG |
| NAGK | TATTTCCAGGTGCCAGATCG | CTGAAGATATAGCGGGAAAGGG |
| USP46 | ATACACCAAGCTGTCTTACCG | ATATAATGCCCACGATTAGGACC |
| ITGB8 | CGTCTCATCTCGCTCTTGATAG | TTCTCTGAAAGTTGGCCTAGTG |
| BMPR2 | GGCTGACTGGAAATAGACTGG | CACAGTCCCTCAAGTTCACAG |
| ZNFX1 | CCGAGGATTGTCATAGTGGAAG | AGATCATACACGTTGGCACTG |
| EPHA5 | AGATTGAGGCAGTGAATGGAG | GCCAAGACAAAGAGATGCTG |
| E2F1 | TCTCCGAGGACACTGACAG | ATCACCATAACCATCTGCTCTG |
| AGPAT1 | AGGACGCAACGTCGAGAAC | GCAGTACCTCCATCATCCCAAG |
| AGPAT2 | GCCGAGTTCTACGCCAAGG | CGAACCAGCCGATGATGCT |
| GPAT1 | GATGTAAGCACACAAGTGAGGA | TCCGACTCATTAGGCTTTCTTTC |
| GPAT2 | TGTGGTCGTCAGGCTTTGG | GGTCCGTTATGCTTCTGTGGA |
| DGAT1 | TATTGCGGCCAATGTCTTTGC | CACTGGAGTGATAGACTCAACCA |
| DGAT2 | ATTGCTGGCTCATCGCTGT | GGGAAAGTAGTCTCGAAAGTAGC |
| FATP2 | TTTCCGCCATCTACACAGTCC | CGTAGGTGAGAGTCTCGTCG |
| FATP3 | CCCTGCTGGAATTAGCGATTT | GGGCGAGGTAGATCACATCTT |
| FATP4 | CTCAGGTCTTGGAGAAGGAAC | ACAGCGGGTCTTTCACAATAG |
| FATP5 | TGGAGGAGATCCTTCCCAAGC | TGGTCCCCGAGGTATAGATGAA |
| FATP6 | CTTCTGTCATGGCTAACAGTTCT | AGGTTTCCGAGGTTGTCTTTTG |
| CPT1B | ATCCTACTCCTATGACCCCG | TCTGCATTGAGACCCAACTG |
| CPT2 | TTGAGTGCTCCAAGTACCATG | GCAAACAAGTGTCGGTCAAAG |
| ACADM | TCATTGTGGAAGCAGATACCC | CAGCTCCGTCACCAATTAAAAC |
| FATP1 | ATGAGGACACAATGGAGCTG | TGACATAGCCATCGAAGCG |
| ACAA1 | GCGGTTCTCAAGGACGTGAAT | GTCTCCGGGATGTCACTCAGA |
| ACSL1 | CCATGAGCTGTTCCGGTATTT | CCGAAGCCCATAAGCGTGTT |
| TBP | CCACTCACAGACTCTCACAAC | CTGCGGTACAATCCCAGAACT |
| B2M | TGCTGTCTCCATGTTTGATGTATCT | TCTCTGCTCCCCACCTCTAAGT |

| | Sequences of the primers for qPCR | |
|---|---|---|
| Mouse Gene | Forward primer | Reverse primer |

*Continued on next page*

| | Sequences of the primers for qPCR | |
|---|---|---|
| Mir20b | GCAAAGTGCTCATAGTGCAGGTAG | TCGCACTTGTCATACACCAG |

*Continued*

| | Sequences of the primers for qPCR | |
|---|---|---|
| *Mir129* | CTTTTTGCGGTCTGGGCTTGC | AGTGCAGGGTCCGAGGTATT |
| *Mir219a* | TGATTGTCCAAACGCAATTC | GAACATGTCTGCGTATCTC |
| *Mir20a* | TGCGCTAAAGTGCTTATAGTGC | CCAGTGCAGGGTCCGAGGTATT |
| *Mir106a* | GATGCTCAAAAAGTGCTTACAGTGCA | TATGGTTGTTCTGCTCTCTGTCTC |
| *Mir106b* | CAAGTACCCACAGTGCGGT | CTCGCTTCGGCAGCACA |
| *Mir17* | TCTAGATCCCGAGGACTG | ATCGTGACCTGAACC |
| *Mir519d* | AGTGCCTCCCTTTAGAG | GAACATGTCTGCGTATCTC |
| *U6* | CTCGCTTCGGCAGCACA | AACGCTTCACGAATTTGCGT |
| *Ppara* | TCAGGGTACCACTACGGAGT | CTTGGCATTCTTCCAAAGCG |
| *Cpt1a* | AGTTCCATGACCCATCTCTGTC | TTCTTCTTCCAGAGTGCAGC |
| *Acox1* | TAACTTCCTCACTCGAAGCCA | AGTTCCATGACCCATCTCTGTC |
| *Srebf1* | GGAGCCATGGATTGCACATT | CTTCCAGAGAGGAGGCCAG |
| *Fasn* | GGAGGTGGTGATAGCCGGTAT | TGGGTAATCCATAGAGCCCAG |
| *Cd36* | GCGACATGATTAATGGCACAG | GATCCGAACACAGCGTAGATAG |
| *Fabp1* | TCTCCGGCAAGTACCAATTG | TTGATGTCCTTCCCTTTCTGG |
| *Tnf* | CCCTCACACTCAGATCATCTTCT | GCTACGACGTGGGCTACAG |
| *Ccl2* | TTAAAAACCTGGATCGGAACCAA | GTTCACCGTAAGCCCAATTT |
| *Il6* | TAGTCCTTCCTACCCCAATTTCC | TTGGTCCTTAGCCACTCCTTC |
| *Il1b* | GCACTACAGGCTCCGAGATGAAC | TTGTCGTTGCTTGGTTCTCCTTGT |
| *Acta2* | GTGAAGAGGAAGACAGCACAG | GCCCATTCCAACCATTACTCC |
| *Col1a1* | CATAAAGGGTCATCGTGGCT | TTGAGTCCGTCTTTGCCAG |
| *Col3a1* | GAAGTCTCTGAAGCTGATGGG | TTGCCTTGCGTGTTTGATATTC |
| *Fn* | CTTTGGCAGTGGTCATTTCAG | ATTCTCCCTTTCCATTCCCG |
| *Timp1* | CTCAAAGACCTATAGTGCTGGC | CAAAGTGACGGCTCTGGTAG |
| *Vim* | CGTCCACACGCACCTACAG | GGGGGATGAGGAATAGAGGCT |
| *E2f1* | TGCAGAACAGATGGTCATAGTG | GGGCACAGGAAAACATCAATG |
| *Rora* | GTGGAGACAAATCGTCAGGAAT | TGGTCCGATCAATCAAACAGTTC |
| *Rorc* | TCCACTACGGGGGTTATCACCT | AGTAGGCCACATTACACTGCT |
| *Thrb* | GGACAAGCACCCATCGTGAAT | CTCTGGTAATTGCTGGTGTGAT |
| *Cpt1b* | GACTTCCGGCTTAGTCGGG | GAATAAGGCGTTTCTTCCAGGA |
| *Cpt2* | CAGCACAGCATCGTACCCA | TCCCAATGCCGTTCTCAAAAT |
| *Acadm* | AACACAACACTCGAAAGCGG | TTCTGCTGTTCCGTCAACTCA |
| *Fatp1* | CTG GGA CTT CCG TGG ACC T | TCT TGC AGA CGA TAC GCA GAA |
| *Acaa1* | TCT CCA GGA CGT GAG GCT AAA | CGC TCA GAA ATT GGG CGA TG |
| *Acsl1* | TGC CAG AGC TGA TTG ACA TTC | GGC ATA CCA GAA GGT GGT GAG |
| *Atgl* | ATGTTCCCGAGGGAGACCAA | GAGGCTCCGTAGATGTGAGTG |
| *Hsl* | GATTTACGCACGATGACACAGT | ACCTGCAAAGACATTAGACAGC |
| *Mgl* | ACCATGCTGTGATGCTCTCTG | CAAACGCCTCGGGGATAACC |
| *Tbp* | ACCCTTCACCAATGACTCCTATG | TGACTGCAGCAAATCGCTTGG |

## Cellular oxygen consumption rate (OCR)

OCR of HepG2 cells were analyzed by Seahorse XF24 extracellular flux analyzer (Seahorse Bioscience, North Billerica, MA) following the manufacturer's instruction. The results were normalized with the protein quantity of each corresponding well.

## Measurement of FA β-oxidation and uptake

FA β-oxidation were measured by the conversion of [9,10-$^3$H(N)]-Palmitic Acid (PerkinElmer, Waltham, MA) to $^3H_2O$. Cells were incubated with 1.25 mCi/L [9,10-$^3$H(N)]-Palmitic Acid with cold palmitic acid in a final concentration of 200 µM for 4 hr. After incubation, medium were recovered and precipitated with an equal volume of 10% tricholoroacetic acid. The supernatant was transferred to new- and capless-microtube and the capless-tube was inserted into D.W-added Scintillation tube and incubated at 60 °C for 12 hr. The capless tube was removed from scintillation tube and measured the CPMA with scincillation counter oil using Tri-carb 2910TR liquid scintillation counter (PerkinElmer).

For FA uptake measurement, cells were incubated with 0.5 µCi/L [9,10-$^3$H(N)]-Palmitic Acid with cold palmitic acid in a final concentration of 200 µM for 2 hours. Uptake was stopped by addition of 200 µM phloretin in 0.1% BSA and lysed in 0.1 N NaOH / 0.03% SDS buffer. The radioactivity of each lysate was counted using Tri-carb 2910TR liquid scintillation counter. The β-oxidation and uptake was normalized to lysate protein concentration determined by BSA assay.

## Metabolites assay

HepG2 cells were transfected with *MIR20B*, anti-*MIR20B*, or MIR NC, respectively. After 48 hr, HepG2 cells (20,000 cells per well) were seeded in 96-well MitoPlate S-1 plates and examined in mitochondrial metabolites activity following the manufacturer's instructions (Biolog, Hayward, CA).

## Mice

All animal experiments were performed according to procedures approved by the Ulsan National Institute of Science and Technology's Institutional Animal Care and Use Committee (UNISTIACUC-19–04). Mice were maintained in a specific pathogen-free animal facility under a 12 hr light/dark cycle at a temperature of 21 °C and allowed free access to water and food. Seven-week-old male C57BL/6 J mice (DBL, Chungbuk, Republic of Korea) were fed a HFD (60% kcal fat, D12492, Research Diets Inc, New Brunswick, NJ) for 12 weeks or a MCD (A02082002BR, Research Diets Inc, New Brunswick, NJ, USA) for 4 weeks. Fenofibrate (100 mg/kg, sc-204751, Santa Cruz biotechnology, Dallas, TX) was administered orally for 4 weeks before mice were sacrificed.

## Hepatocyte isolation

Briefly, mice were anesthetized with isoflurane, and 24-gauge needle was inserted into the portal vein. Then the inferior vena cava was cut, and the mouse liver was perfused sequentially with solution I (142 µM NaCl, 6.7 µM KCl, 10 µM HEPES, and 2.5 mM EGTA), and solution II (66.7 mM NaCl, 6.7 mM KCl, 50 mM HEPES, 4.8 mM CaCl$_2$·2H$_2$O, and 0.01% Type IV collagenase (Sigma- Aldrich, St. Louis, MO)). After digestion, the liver was disrupted over a 70 µm cell strainer, and cell suspension was spun at 50 x g for 5 min at 4 °C. The supernatant was gently aspirated and the cells were resuspended in M199 with EBSS (M199/EBSS) medium and gently mixed with equal volume of Percoll working solution (48.6% Percoll). The cell suspension was spun at 100 x g for 5 min at 4 °C, and the pellet washed once with M199/EBSS. After viable cells were counted with trypan blue, the isolated hepatocytes were seed in M199/EBSS medium supplemented with 10% FBS, 1% penicillin/streptomycin, and 10 nM dexamethasone.

## Production of AAV

The *Mir20b* and anti-*Mir20b* were cloned into the pOTTC385-pAAV CMV-IE IRES EGFP vector (Addgene plasmid # 102936) (*Nelson et al., 2019*) and co-transfected with pAAV-DJ vector and pAAV-Helper vector into HEK 293T cells to generate recombinant adeno-associated virus expressing *Mir20b* or anti-*Mir20b* according to the manufacturer's protocol (Cell Biolabs, San Diego, CA). The AAVs were purified with AAVpro Purification Kit (Takara Bio, Shiga, Japan). When feeding HFD was started, purified AAV-*Mir20b*, AAV-anti-*Mir20b*, or AAV-Control (1 × 10$^{10}$ PFU) was injected into mice

*via* tail-vein. AAV-Control or AAV-anti-*Mir20b* was injected into mice *via* tail-vein before the initiation of MCD diet.

## Lentivirus preparation, transduction, and stable cell line generation

All lentivirus-based shRNA clones used for making the viral transduction particles were purchased from Sigma-Aldrich. pLKO.1-Puro vector targeting human *PPARA* or non-target vector as a control was used. In short, 293T cells were plated at a density of $12 \times 10^6$ 293 T cells in a 150 mm dish containing 20 mL of media. After 24 hr, the cells were overlaid with a complex containing a three-plasmid system (pLKO.1 shRNA, pCMV-VSVG, and pCMV-Δ8.9) at the ratio of 4:2:3 using lipofectamine (*Jang et al., 2020*). The supernatant was collected starting from 48 to 72 hr post-transfection. The virus particles were concentrated by ultracentrifugation at 25,000 rpm for 1 hr 30 min with a Beckman ultracentrifuge using the SW28 rotor, resuspended in PBS, and stored until use at −70 °C. Harvested virus was used to transduce target cells for 24 hr. Stable cell lines were selected in the presence of 3 μg/mL puromycin for 5 days. To minimize batch-to-batch differences, we compared only cells made from the same batch of cells.

| Name | Sequence |
|------|----------|
| *shPPARA* #1 | GAACAGAAACAAATGCCAGTA |
| *shPPARA* #2 | GTAGCGTATGGAAATGGGTTT |

## Metabolic analysis

Mice were fasted overnight (18 hr) before intraperitoneal injection of D-glucose (2 g/kg body weight) for glucose tolerance test. For insulin tolerance test, mice were fasted for 4 hr before intraperitoneal injection of insulin (0.75 U/kg body weight). Every glucose was examined with tail-vein blood at indicated intervals after injection using a glucometer. For analyzing metabolic parameters, insulin (90080, Crystal Chem, Elk Grove Village, IL), ALT (K752, Biovision Inc, Milpitas, CA), AST (K753, Biovision Inc), cholesterol (K603, BioVision Inc), TG (10010303, Cayman Chemical, Ann Arbor, MI), and FFA (BM-FFA-100, BioMax, Republic of Korea) were determined. Body composition of mice was measured using an EchoMRI100V, quantitative nuclear resonance system (Echo Medical Systems, Houston, TX).

## Histological analysis

Liver tissues were isolated from mice and immediately fixed with 4% formalin (Sigma- Aldrich, St. Louis, MO). Histological changes of lipid droplets were examined by H&E staining and Oil Red O staining. As counterstain, Mayer's hematoxylin was used for every slide. Liver fibrosis was further examined by Sirius red with liver section. Images were obtained with Olympus BX53 microscope and DP26 camera. NAFLD activity score (NAS) and fibrosis score were measured by a pathologist according to the NASH Clinical Research Network scoring system (*Brunt et al., 2011*). For statistical analysis of fibrosis score, each grade (0~4) was calculated by giving a score from 1 to 6 points.

## Statistical analysis

All data are represented as mean ± SEM. Statistically significant differences were assessed by the Student's t-test. Statistical analyses and Pearson correlation analyses were performed using Microsoft Excel or GraphPad Prism 9.3.0 (RRID:SCR_002798). All the significance are expressed as $*p < 0.05$, $**p < 0.01$, $***p < 0.001$, $^\#p < 0.05$, $^{\#\#}p < 0.01$, $^{\#\#\#}p < 0.001$, $^\$p < 0.05$, $^{\$\$}p < 0.01$, and $^{\$\$\$}p < 0.001$. ns, not significant.

# Results

## *MIR20B* significantly increases in the livers of dietary obese mice and human

We constructed a regulatory network of NRs that were differentially expressed in NAFLD patients (*Hoang et al., 2019*) and microRNA targeting NRs based on miRNA target prediction (*Agarwal et al., 2015*), to identify the correlation between NR and microRNA in the development of NAFLD. As shown in *Figure 1A*, the top ten miRNAs were found to be highly correlated with the modulation

of NR expression in NAFLD. To further prioritize key miRNAs, we assessed the expression of miRNA candidates in NASH, simple steatosis, and normal patient liver samples (*Figure 1B*, *Figure 1—figure supplement 1*). Among the selected miRNAs, the expression of *MIR20B* was significantly increased in simple steatosis and NASH compared to that in normal individuals, and the extent of increase in NASH was higher than that in simple steatosis. In particular, the expression of *MIR20B* showed a significant correlation with the BMI, AST, ALT, and fasting glucose during NAFLD progression (*Figure 1—figure supplement 2*).

*MIR20B* expression was also most significantly increased in both oleic acid (OA)-treated HepG2 and Huh-7 cells (*Figure 1C and D*, *Figure 1—figure supplement 4A, B*). Moreover, the expression of *MIR20B* was significantly upregulated in the fatty livers of high-fat diet (HFD)-fed mice, *Lep$^{ob/ob}$* mice, and methionine-deficient diet (MCD)-fed mice compared to that in the liver of normal chow diet (NCD)-fed wild mice (*Figure 1E–G*, *Figure 1—figure supplement 3*, and *Figure 1—figure supplement 4C*). Together, our results indicate that *MIR20B* expression is increased in NAFLD and is highly associated with the regulation of NRs in NAFLD.

## *PPARA* is a direct target of *MIR20B*

Next, we characterized the physiological roles of *MIR20B* in NAFLD. Oil Red O staining showed that *MIR20B* expression increased intracellular lipid content, and this lipid accumulation was increased with OA treatment in HepG2 cells (*Figure 2A*). As expected, overexpression of *MIR20B* significantly upregulated TG content and cholesterol in OA-treated HepG2 cells (*Figure 2B and C*). To further investigate the functions and targets of *MIR20B*, HepG2 cells were transfected with *MIR20B* and two separate RNA-Seq libraries of non-targeting control (MIR NC) and forced *MIR20B* expression (*MIR20B*) were constructed. The six RNA-seq samples were clearly separated between the forced *MIR20B* expression and control conditions, implying a significant impact on gene expression in HepG2 cells (*Figure 2D*, *Figure 2—figure supplement 1A*). The NR metapathway was detected as the most significant pathways (adjusted p-value = 2.47-E8) by the gene-set enrichment analysis (GSEA) (*Subramanian et al., 2005*), indicating that NRs and related genes are the major targets of *MIR20B* (*Figure 2E and F*). The heatmap of RNA-seq data for the NR transcription pathway is shown in *Figure 2G*. Five NRs (*PPARA*, *RORA*, *RORC*, *THRB*, and *NRBP1*) were downregulated (adjusted p-value ≤ 0.05), and *PPARA* was the most significantly downregulated NR (adjusted p-*value* = 3.25-E5). Furthermore, GSEA showed that PPARA pathways and PPAR signaling pathways were significantly decreased in *MIR20B* overexpressed cells (*Figure 2—figure supplement 1B*, C). We validated the expression of these NRs and the *PPARA*, *RORC*, *THRB*, and *NRBP1* expression was decreased by *MIR20B* in human liver cells and mouse primary hepatocytes. Consistent with the RNA-seq data, the expression change of *PPARA* at both the protein and mRNA levels with *MIR20B* transfection was the most distinct compared to the control (*Figure 2H1*). Moreover, among candidate targets from RNA-seq analysis, only *PPARA* was selected as an overlapped predicted target of *MIR20B* between various miRNA target prediction programs, including miRDB, picTAR, TargetSCAN, and miRmap (*Figure 2J*, *Figure 2—figure supplement 2*). While the expression of other candidates, including *RORA*, *RORC*, and *THRB*, remained unchanged, that of *PPARA* significantly decreased as NAFLD progressed to steatosis and NASH, and showed a negative correlation with that of *MIR20B* (*Figure 2—figure supplement 3*). In addition, *PPARA* decreased in diverse diets-induced NAFLD conditions with *MIR20B* (*Figure 1C–E*, *Figure 2—figure supplement 4*).

Notably, the 3'UTR of *PPARA* mRNA includes *MIR20B* binding sites that are well conserved between humans and mice, suggesting that *MIR20B* may have a direct inhibitory effect on *PPARA* expression (*Figure 2K*). Using a luciferase reporter construct including the 3'UTR of *PPARA*, we revealed that *MIR20B* suppressed the luciferase activity in both HepG2 and Huh-7 cells in a dose-dependent manner. Furthermore, we built the mutant construct of the predicted *MIR20B*-binding sites within the 3'UTR of *PPARA*; the inhibitory effect of *MIR20B* on luciferase activity was completely blunted (*Figure 2L and M*). To further validate the functional role of 3'UTR regulated by *MIR20B*, we introduced the wild-type *PPARA* open-reading frame (ORF) followed by either the wild type (WT) or mutant (Mut) 3'UTR of *PPARA* in HepG2 cells (*Figure 2—figure supplement 5A*). As shown in *Figure 2—figure supplement 5C*, *MIR20B* significantly suppressed the expression of *PPARA* and its target genes in *PPARA*-3'UTR WT expressing cells. Furthermore, Oil Red O staining showed that *MIR20B* increased the intracellular lipid content in these cells (*Figure 2—figure supplement 5B*). However, *MIR20B* did

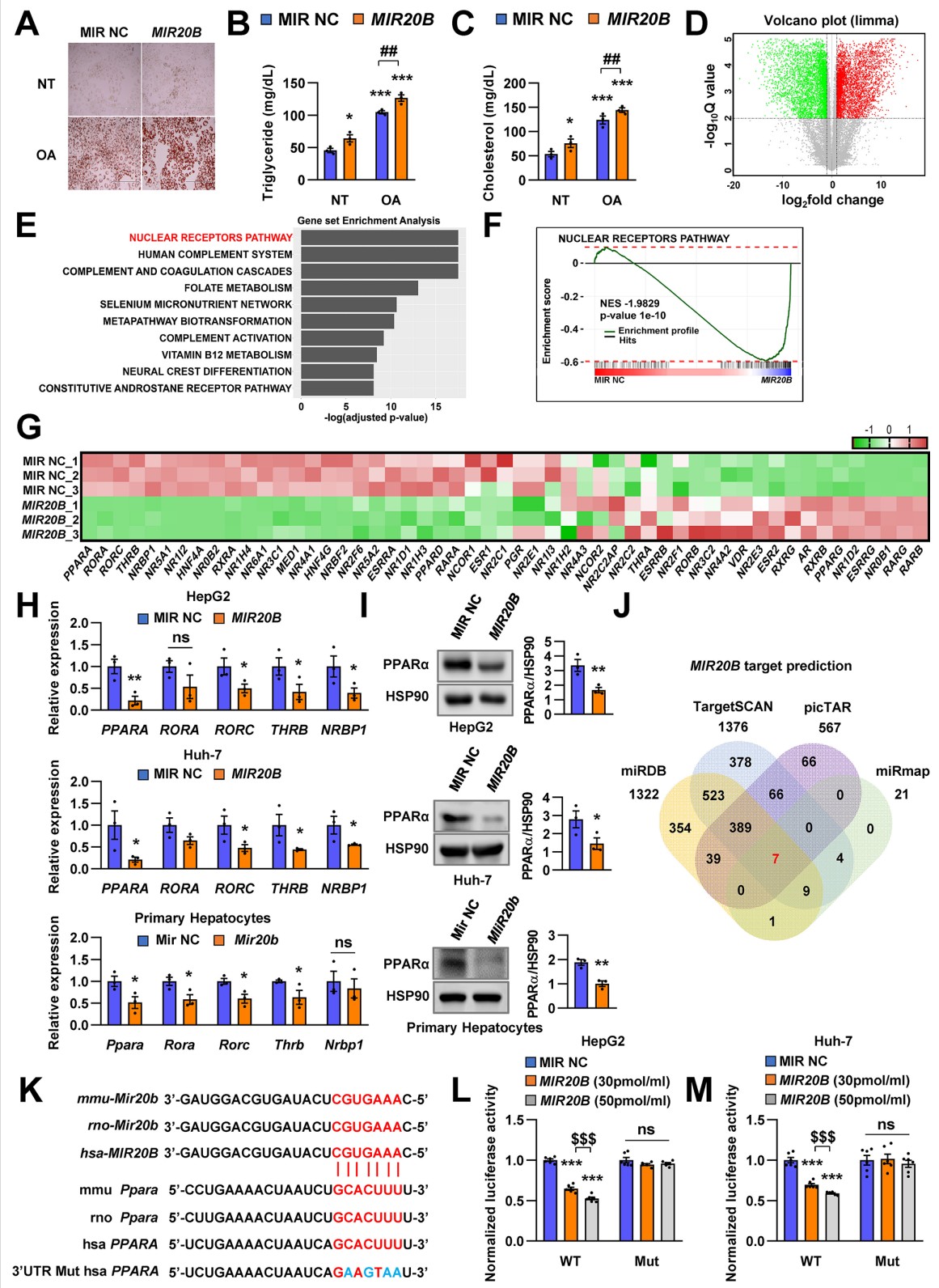

**Figure 2.** *PPARA* is a direct target of *MIR20B*. Overexpressed *MIR20B* induces hepatic lipid accumulation in HepG2 cells treated with OA (1 mM). Oil Red O staining showed intracellular lipid accumulation in HepG2 cells. Scale bar is 400 μm (**A**). TG (**B**) and Cholesterol levels (**C**) were examined in OA-treated HepG2 cells transfected with MIR NC or *MIR20B*. RNA-seq was performed on sample from HepG2 cell with or without overexpression of *MIR20B*.Volcano plot of the gene expressions (log2 fold change) compared to the negative control from RNA-seq analysis (**D**). Top ranked GSEA in

*Figure 2 continued on next page*

*Figure 2 continued*

overexpressed *MIR20B* compared to MIR NC in HepG2 cells (**E**). The primary ranked enrichment plot of nuclear receptors pathway (**F**). Heatmap of the genes in NR pathway upon *MIR20B* overexpression compared to control (**G**). Expression of primary ranked nuclear receptors pathway genes from RNA-seq analysis in HepG2 cells, Huh-7 cells, and primary hepatocytes transfected with *MIR20B* were normalized to each cells transfected with MIR NC (**H**). Western blot analysis of PPARα on MIR NC or *MIR20B* transfected cells. The intensity of PPARα blot was normalized to that of HSP90 (**I**). Venn diagram of predicted targets for *MIR20B* in four major database system (**J**). Graphic image of the conserved binding motifs of *MIR20B* within 3'UTR mRNA of *PPARA* (**K**). Luciferase activities of *MIR20B*-transfected HepG2 cells and Huh-7 cells containing the luciferase reporter DNA constructs for wild-type or mutated 3'UTR of *PPARA* were normalized to those or MIR NC-transfected cells (**L, M**). Values represent means ± SEM (n = 3). *$p < 0.05$, **$p < 0.01$, ***$p < 0.001$ *vs* MIR NC. ##$p < 0.01$ *vs* MIR NC+ OA. \$\$\$ $< 0.001$ *vs MIR20B* (30 pmol/ml). Raw data can be found in a Source Data file named '***Figure 2—source data 1***'.

The online version of this article includes the following source data and figure supplement(s) for figure 2:

**Source data 1.** *PPARA* is a direct target of *MIR20B*.

**Figure supplement 1.** Analysis of PPARα related pathway in RNA-seq of *MIR20B* overexpressed HepG2 cells.

**Figure supplement 2.** *PPARA* is the primary target of the overlapped candidates.

**Figure supplement 2—source data 1.** *PPARA* is the primary target of the overlapped candidates.

**Figure supplement 3.** Correlation between the expression of nuclear receptors and *MIR20B* during NAFLD progression.

**Figure supplement 3—source data 1.** Correlation between the expression of nuclear receptors and *MIR20B* during NAFLD progression.

**Figure supplement 4.** The expression of *PPARA* is regulated both in human and mice.

**Figure supplement 4—source data 1.** The expression of *PPARA* is regulated both in human and mice.

**Figure supplement 5.** *PPARA*-3'UTR mutant blocks the effects of *MIR20B* on lipid metabolism.

**Figure supplement 5—source data 1.** *PPARA*-3'UTR mutant blocks the effects of *MIR20B* on lipid metabolism.

not have an effect on either the expression of *PPARA* and its target genes or the intracellular lipid content in *PPARA*-3'UTR Mut expressing cells (***Figure 2—figure supplement 5B-D***). Taken together, these results indicate that *MIR20B* inhibits the expression of *PPARA* by interacting with its 3'UTR.

## The effects of *MIR20B* are mediated by PPARα

Next, we confirmed the effect of *MIR20B* on function of PPARα. Since PPARα is a master regulator of lipid metabolism such as FA utilization and oxidation, we investigated the effects of *MIR20B* on expression of genes involved in lipid metabolism. Transfection of *MIR20B* into HepG2 cells reduced the expression and activity of PPARα, but co-transfected *PPARA* expression vector restored them (***Figure 3A and B***). Overexpression of *MIR20B* reduced the expression of genes involved in FA β-oxidation and FA uptake, including *CPT1A, ACOX1, CD36,* and *FABP1*, but not de novo lipogenesis such as *SREBF1* and *FASN*, which was significantly restored by the forced expression of PPARα (***Figure 3C***). Next, we tested whether anti-*MIR20B* enhanced activity of PPARα and the effects of anti-*MIR20B* was specific to PPARα. Ectopic expression of anti-*MIR20B* increased the expression and activity of PPARα (***Figure 3D and E***), and consequently enhanced the expression of genes involved in FA utilization and oxidation (***Figure 3F***). *PPARA*-targeting siRNA (*siPPARA*) suppressed the expression of *PPARA* and its activity (***Figure 3D and E***). The increased expression of genes by anti-*MIR20B* was also suppressed by *siPPARA* (***Figure 3F***). In addition, fenofibrate, a PPARα agonist, increased the expression of PPARα and its transcriptional activity in HepG2 cells transfected with *MIR20B*, but could not restore as much on its own effects (***Figure 3G and H***). Interestingly, fenofibrate treatment increased the expression of genes involved in FA β-oxidation and FA uptake which are regulated by PPARα, but could not overcome the inhibitory effect of *MIR20B* (***Figure 3I***). Taken together, these results indicate that *MIR20B* affects lipid metabolism through direct inhibition of *PPARA*.

## *MIR20B* regulates fatty acid metabolism

We investigated the effects of *MIR20B* on lipid metabolism to reveal the functional contribution of increased *MIR20B* to NAFLD. Unlike in HepG2 cells (***Figure 2A–C***), *MIR20B* alone did not induce lipid accumulation in primary hepatocytes without OA treatment, but *MIR20B* significantly increased lipid accumulation in the presence of OA (***Figure 4—figure supplement 2A-D***). The mRNA levels of genes involved in FA β-oxidation, FA uptake, including *CPT1A, ACOX1, CD36,* and *FABP1*, and PPARα target genes were decreased by overexpression of *MIR20B* compared to the control in HepG2

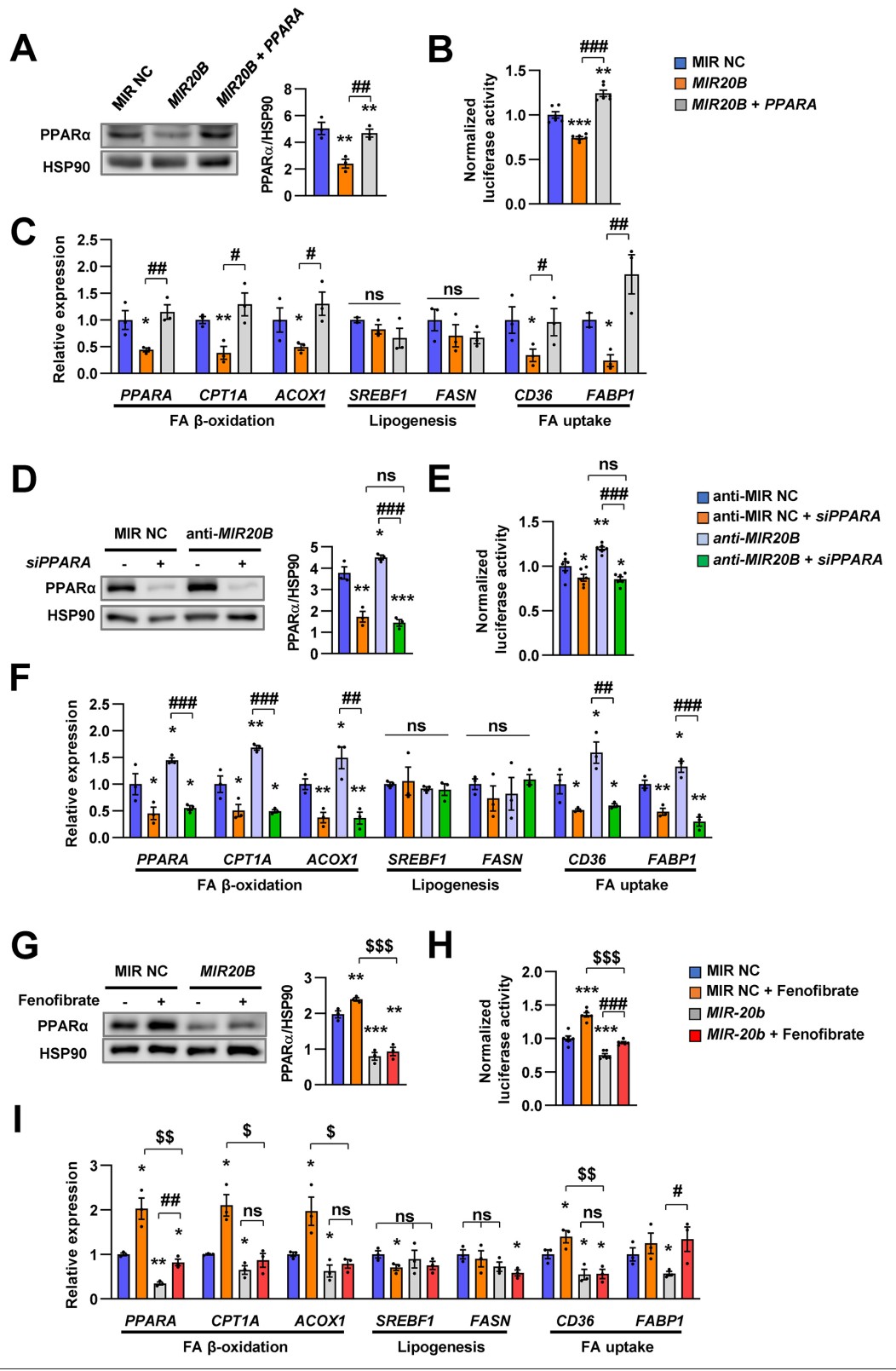

**Figure 3.** The effects of *MIR20B* are mediated by PPARα. HepG2 cells transfected with indicated miRNA, siRNA, or *PPARA* expression vector. Western blot analysis of PPARα. The intensity of PPARα blot was normalized to that of HSP90 (**A, D**). Luciferase activity using the luciferase reporter DNA constructs containing PPRE (*PPAR* response element) was transfected in HepG2 cells. Luciferase activity was normalized to renilla activity (**B, E**). mRNA level

*Figure 3 continued on next page*

*Figure 3 continued*

of genes involved in FA β-oxidation, lipogenesis, and FA uptake was analyzed by real-time qPCR (**C and F**). After transfected with MIR NC or *MIR20B*, HepG2 cells were treated with fenofibrate (100 μM). Protein level of PPARα was analyzed by western blot and normalized to that of HSP90 (**G**). The transcriptional activity of PPARα was measured (**H**). Genes involved in of FA β-oxidation, lipogenesis and FA uptake were determined by real-time qPCR (**I**). Relative values are normalized to MIR NC. Values represent means ± SEM (n = 3). ns, not significant. *p < 0.05, **p < 0.01, ***p < 0.001 *vs* MIR NC. #p < 0.05, ##p < 0.01, ###p < 0.001 *vs MIR20B* or anti-*MIR20B*. $p < 0.05, $$p < 0.01, $$$p < 0.001 *vs* MIR NC+ fenofibrate. Raw data can be found in a Source Data file named '***Figure 3—source data 1***'.

The online version of this article includes the following source data for figure 3:

**Source data 1.** The effects of *MIR20B* are mediated by PPARα.

cells and primary hepatocytes (***Figure 4A and B***, ***Figure 4—figure supplement 1A***), whereas the genes associated with de novo lipogenesis and ketogenesis were not affected by *MIR20B* (***Figure 4B***, ***Figure 4—figure supplements 2E***; ***Figure 4—figure supplement 3A, B***). Interestingly, *DGAT1*, rate limiting enzyme for TG synthesis, was increased in HepG2 cells upon OA treatment (***Figure 4C***). *MIR20B* overexpressed HepG2 cells showed reduced levels of palmitoyl-carnitine, a substrate of β-oxidation, and acetyl-CoA, a product of β-oxidation. Subsequently, TCA cycle intermediate levels, including citrate and succinate, also decreased (***Figure 4D***).

Enforced expression of *MIR20B* in HepG2 cells under both basal and OA treatments decreased the expression of *PPARGC1A* and *SIRT1*, which are involved in mitochondrial biogenesis (***Figure 4E***). The copy number of two mtDNA genes, *VIPR1* and *MT-ATP6*, was decreased by *MIR20B* overexpression following OA treatment (***Figure 4F***). Consistently, mitochondrial function that was analyzed *via* OCR (oxygen consumption rate) was reduced by *MIR20B* under both basal and OA treatment conditions compared to the control (***Figure 4G***). In particular, the basal respiration and maximal respiratory capacity were significantly suppressed by *MIR20B* (***Figure 4H***). Furthermore, the level of ATP production, FA uptake, and FA oxidation was reduced in *MIR20B* overexpressed cells compared with that in the control under both basal and OA-treated conditions (***Figure 4I–K***).

To further clarify the role of *MIR20B* in hepatic fatty acid metabolism, anti-*MIR20B* was delivered into OA-treated HepG2 cells and primary hepatocytes, and which reduced the expression of *MIR20B* (***Figure 4L***, ***Figure 4—figure supplement 4G***). Oil red O staining showed that anti-*MIR20B* remarkably decreased intracellular lipid accumulation upon OA treatment (***Figure 4—figure supplement 4A, D***). As expected, *MIR20B* inhibition reduced the levels of both TG and cholesterol under OA conditions compared to the control (***Figure 4—figure supplement 4B***, C, E, and F). Lipid consumption-associated genes and PPARα target genes, not de novo lipogenic genes, were significantly upregulated in *MIR20B* inhibited HepG2 cells and primary hepatocytes compared to those in the control under both basal and OA conditions (***Figure 4M***, ***Figure 4—figure supplements 1B***; ***4H and I***). TG synthesis enzymes, such *GPAT1* and *AGPAT1*, were decreased by anti-*MIR20B* in HepG2 cells under OA-treated condition, however *DGAT1* and *DGAT2* were reduced under both basal and OA-treated conditions (***Figure 4N***). Inhibition of *MIR20B* increased the levels of palmitoyl-carnitine, acetyl-CoA, and TCA cycle intermediates (***Figure 4O***), whereas ketogenesis was not affected (***Figure 4—figure supplement 3***). Furthermore, the expression of mitochondrial biogenesis genes (***Figure 4P***) and the copy number of mitochondrial DNA genes were increased in both basal and OA conditions (***Figure 4Q***). Consequently, anti-*MIR20B* treatment significantly upregulated the mitochondrial activity, FA uptake, and FA oxidation (***Figure 4R–V***). Taken together, these results demonstrated that *MIR20B* contributes to hepatic lipid accumulation by controlling lipid oxidation, mitochondrial function and TG synthesis through changes in gene expression, further contributing to the progression of NAFLD.

## *Mir20b* promotes hepatic steatosis in HFD-fed mice

To confirm the in vivo roles of *Mir20b* in obesity model mice, we introduced *Mir20b* using an adenovirus-associated vector (AAV), referred to as AAV-*Mir20b*, into C57BL/6 mice that had been fed a normal chow diet (NCD) or a high-fat diet (HFD). Administration of AAV- *Mir20b* led to high expression levels of *Mir20b* in the livers of NCD- and HFD-fed mice compared to AAV-Control injection (***Figure 5A***). However, the expression level of *Mir20b* was not changed in adipose tissues including white and brown adipose tissues compared to NCD- or HFD-fed AAV-Control injected mice, respectively

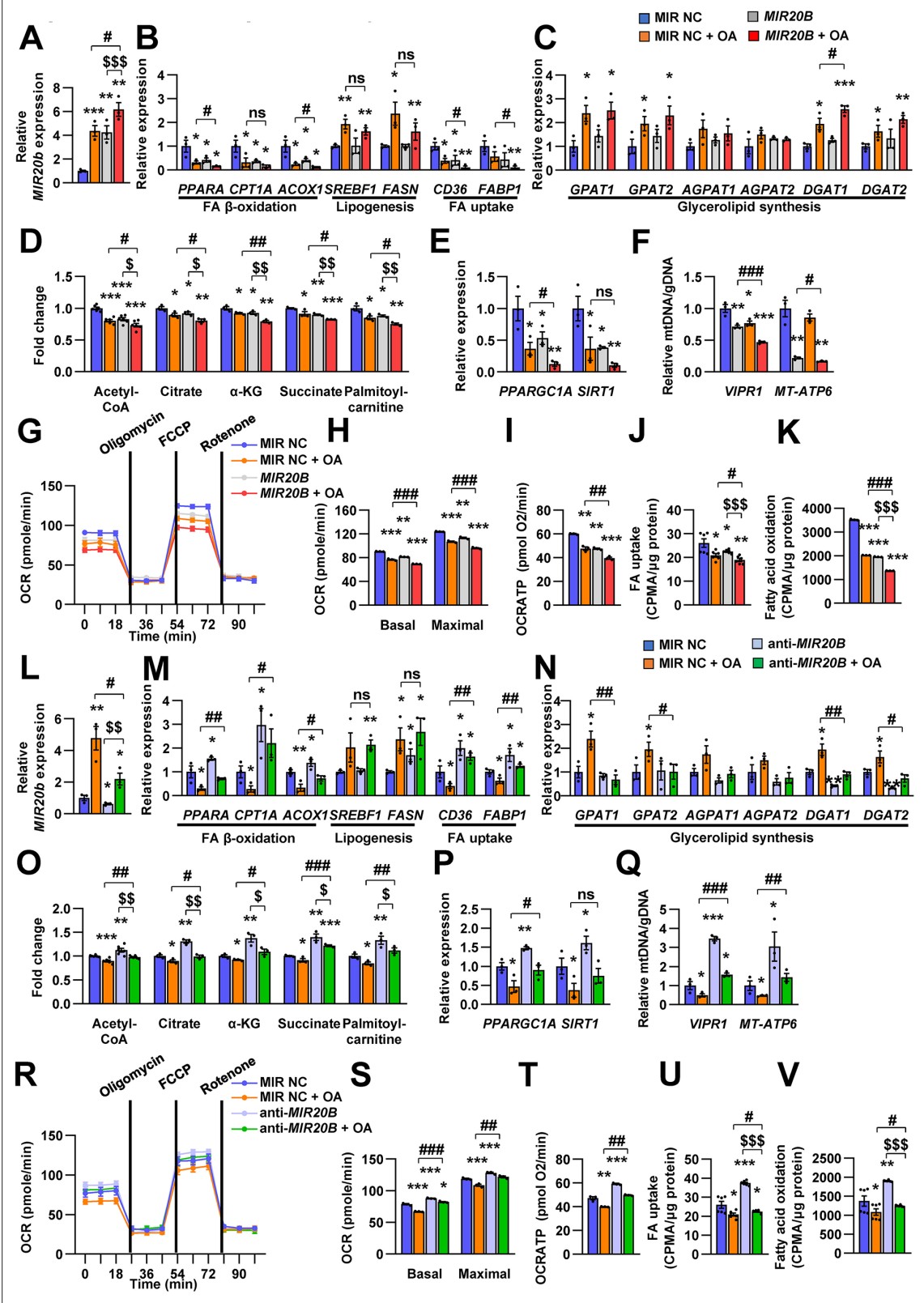

**Figure 4.** *MIR20B* regulates fatty acid metabolism. HepG2 cells were transfected with *MIR20B* or anti-*MIR20B* and treated with OA for 24 hr. The expression of *MIR20B* (**A, L**) and genes related to FA β-oxidation, lipogenesis, FA uptake (**B, M**) and glycerolipid synthesis (**C, N**) were measured by quantitative RT-PCR. Representative mitochondrial metabolites were measured in HepG2 cells (**D, O**). The expression of genes related to mitochondrial biogenesis (**E, P**) were measured by quantitative RT-PCR. The mitochondrial copy of VIPR1 and MT-ATP6 were determined (**F, Q**). OCR (**G, R**), basal

*Figure 4 continued on next page*

*Figure 4 continued*

and maximal OCR (**H, S**), and ATP levels (**I, T**) were measured in HepG2 cells. FA uptake (**J, U**) and β-oxidation (**K, V**) activity were measured using [9,10-$^3$H(N)]-Palmitic Acid and normalized to the total protein content. Relative values are normalized to MIR NC. Values represent means ± SEM (n = 3–5). ns, not significant. *p < 0.05, **p < 0.01, ***p < 0.001 vs MIR NC. #p < 0.05, ##p < 0.01, ###p < 0.001 vs MIR NC+ OA. $p < 0.05, $$p < 0.01, $$$p < 0.001 *vs MIR20B* or anti-*MIR20B*, respectively. Raw data can be found in a Source Data file named '*Figure 4—source data 1*'.

The online version of this article includes the following source data and figure supplement(s) for figure 4:

**Source data 1.** *MIR20B* regulates fatty acid metabolism.

**Figure supplement 1.** The expression of PPARα target genes is regulated by *MIR20B*.

**Figure supplement 1—source data 1.** The expression of PPARα target genes is regulated by *MIR20B*.

**Figure supplement 2.** *MIR20B* enhances lipid accumulation in primary hepatocytes under OA-treatment.

**Figure supplement 2—source data 1.** *MIR20B* enhances lipid accumulation in primary hepatocytes under OA-treatment.

**Figure supplement 3.** Ketogenesis is not regulated by *MIR20B* in HepG2 cells.

**Figure supplement 3—source data 1.** Ketogenesis is not regulated by *MIR20B* in HepG2 cells.

**Figure supplement 4.** Inhibition of *MIR20B* alleviates lipid accumulation in HepG2 cells.

**Figure supplement 4—source data 1.** Inhibition of *MIR20B* alleviates lipid accumulation in HepG2 cells.

**Figure supplement 5.** The expression of genes associated with FA uptake by *MIR20B* in HepG2 cells.

**Figure supplement 5—source data 1.** The expression of genes associated with FA uptake by *MIR20B* in HepG2 cells.

**Figure supplement 6.** The expression of *MIR20B* by OA in HepG2 cells.

**Figure supplement 6—source data 1.** The expression of *MIR20B* by OA in HepG2 cells.

(*Figure 5—figure supplement 1*). Consequently, AAV-*Mir20b* injected mice exhibited a reduction in the expression of *Ppara* and PPARα target genes compared with AAV-Control injected mice on both NCD and HFD (*Figure 5B and R*, and *Figure 5—figure supplement 2*).

Alterations in body weight were not detected in NCD-fed mice after AAV-*Mir20b* administration; however, AAV-*Mir20b* led to a significant increase in the body weight of HFD-induced obese mice (*Figure 5C*). The ratio of fat mass to body weight in AAV-*Mir20b* administration HFD-fed mice was higher than that in AAV-Control treated mice (*Figure 5D* and *Figure 5—figure supplement 3*); however, the ratio of lean mass to body weight was decreased (*Figure 5E*). Consistently, AAV-*Mir20b* administration increased liver weight, steatosis, and NAFLD activity score (NAS) in HFD-fed mice (*Figure 5F–H*). The hepatic TG level, serum activities of aspartate aminotransferase (AST) and alanine aminotransferase (ALT), markers of liver injury, were significantly increased with AAV-*Mir20b* administration compared with AAV-Control administration in HFD-fed mice (*Figure 5I–K*).

Additionally, we observed that delivery of AAV-*Mir20b* to HFD-fed mice significantly impaired glucose tolerance and insulin sensitivity compared to the AAV-Control (*Figure 5L and M*). Fasting glucose, insulin, and homeostasis model assessment of insulin resistance (HOMA-IR) levels were also increased in AAV-*Mir20b* administrated HFD-fed mice (*Figure 5N–P*). We observed that genes involved in FA β-oxidation and FA uptake pathways were downregulated by AAV-*Mir20b* compared to AAV-Control in both NCD- and HFD-fed mice, whereas lipogenesis genes were not altered in AAV-*Mir20b* administrated mice (*Figure 5Q*). These results suggest that *Mir20b* could aggravate NAFLD by dysregulating lipid metabolism in a HFD-induced obesity model.

## Inhibition of *Mir20b* alleviates hepatic steatosis in HFD-fed mice

Next, we introduced anti-*Mir20b* into HFD-fed mice. Administration of AAV-anti-*Mir20b* led to decrease of *Mir20b* in the livers of NCD- and HFD-fed mice compared to AAV-Control injection (*Figure 6A* and *Figure 6—figure supplement 1*). AAV-anti-*Mir20b* significantly increased expression of *Ppara* and its target genes in the livers of both NCD- and HFD-fed mice (*Figure 6B and R* and *Figure 6—figure supplement 2*). Administration of AAV-anti-*Mir20b* in HFD-fed mice reduced the body weight compared to that of AAV-Control administrated mice (*Figure 6C*). We further determined that alterations in body weight were associated with fat mass loss and reduced trend of epididymal white adipose tissue (*Figure 6D* and *Figure 6—figure supplement 3*). While the ratio of lean mass to body weight of AAV-anti-*Mir20b* administrated HFD-fed mice was increased, the lean mass was comparable to that of the control (*Figure 6E*). We next observed that AAV-anti-*Mir20b* administration reduced liver weight and hepatic steatosis in HFD-fed mice than in AAV-Control mice (*Figure 6F*). H&E and

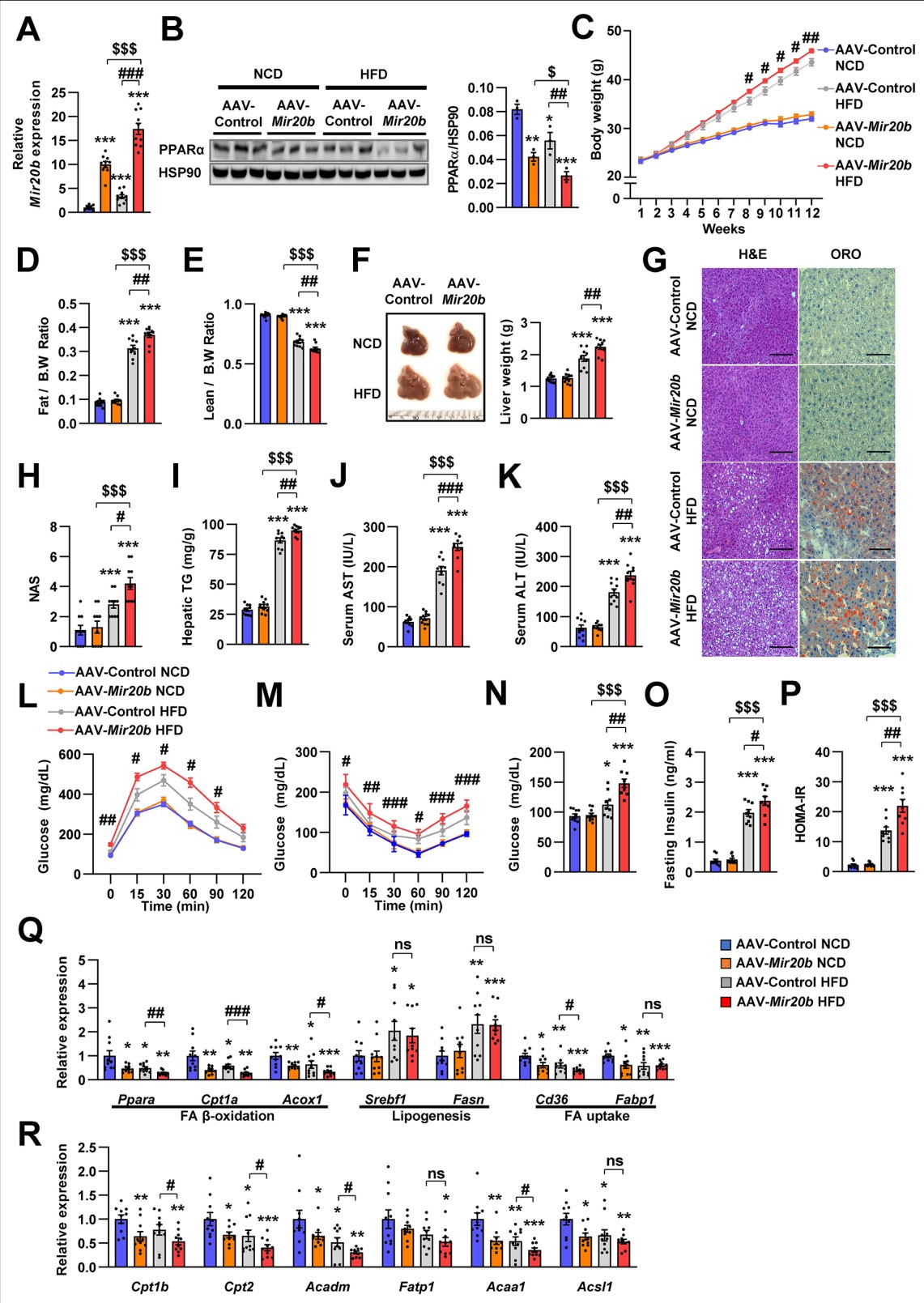

**Figure 5.** *Mir20b* promotes hepatic steatosis in HFD-fed mice. C57BL/6 J mice were fed on normal chow diet (NCD, n = 10 per group) or high fat diet (HFD, n = 10 per group) for 12 weeks and administered with indicated AAVs. Hepatic expression of *Mir20b* (**A**) and PPARα (**B**), body weight (**C**), the ratio of fat mass to body weight (**D**), the ratio of lean mass to body weight (**E**), representative images and weight of liver (**F**), representative images of H&E staining and Oil Red O staining of liver slides (**G**), NAFLD activity score (NAS) (**H**), hepatic TG (**I**), serum AST (**J**), serum ALT (**K**), glucose tolerance

*Figure 5 continued on next page*

*Figure 5 continued*

(**L**), insulin tolerance (**M**), fasting glucose (**N**), fasting insulin (**O**), and HOMA-IR (**P**) were analyzed in indicated mice. Genes related to FA β-oxidation, lipogenesis and FA uptake (**Q**) and PPARα target genes (**R**) were determined by quantitative RT-PCR. Relative values are normalized to AAV-Control NCD. The intensity of PPARα blot was normalized to that of HSP90. Values represent means ± SEM (n = 10). ns, not significant. *p < 0.05, **p < 0.01, ***p < 0.001 *vs* AAV-Control NCD. #p < 0.05, ##p < 0.01, ###p < 0.001 *vs* AAV-Control HFD. $p < 0.05, $$$p < 0.001 *vs* AAV-*Mir20b* NCD. Scale bar is 100 µm. Raw data can be found in a Source Data file named '*Figure 5—source data 1*'.

The online version of this article includes the following source data and figure supplement(s) for figure 5:

**Source data 1.** *Mir20b* promotes hepatic steatosis in HFD-fed mice.

**Figure supplement 1.** The expression of *Mir20b* in adipose tissues of AAV injected mice.

**Figure supplement 1—source data 1.** The expression of *Mir20b* in adipose tissues of AAV injected mice.

**Figure supplement 2.** *Mir20b* significantly regulates *Ppara* in HFD-fed mice.

**Figure supplement 2—source data 1.** *Mir20b* significantly regulates *Ppara* in HFD-fed mice.

**Figure supplement 3.** The weight of adipose tissues of AAV injected mice.

**Figure supplement 3—source data 1.** The weight of adipose tissues of AAV injected mice.

**Figure supplement 4.** The expression of genes involved in lipolysis in adipose tissue and serum FFA in AAV injected mice.

**Figure supplement 4—source data 1.** The expression of genes involved in lipolysis in adipose tissue and serum FFA in AAV injected mice.

Oil Red O staining demonstrated that delivery of AAV-anti-*Mir20b* significantly attenuated the size and number of lipid droplets in the liver and NAS compared to AAV-Control administration in HFD-fed mice (*Figure 6G and H*). In accordance with histological changes, metabolic parameters were reduced in AAV-anti-*Mir20b* administered mice compared with the AAV-Control administrated mice (*Figure 6I–K*). Furthermore, AAV-anti-*Mir20b* significantly improved glucose tolerance (*Figure 6L*) and insulin sensitivity (*Figure 6M*) compared to the AAV-Control in HFD-fed mice. Consistently, we determined that fasting glucose, fasting insulin, and HOMA-IR levels were decreased by AAV-anti-*Mir20b* (*Figure 6N–P*). Delivery of AAV-anti-*Mir20b* increased the expression of genes associated with FA β-oxidation and FA uptake compared with the administration of AAV-Control (*Figure 6Q*). Together, these results suggest that suppression of *Mir20b* could ameliorate NAFLD by recovering lipid metabolism in a HFD-induced obesity model.

## The effects of fenofibrate are limited in *Mir20b*-introduced mice

Next, we tested whether NAFLD treatment with fenofibrate was affected by *Mir20b* expression in vivo. Administration of AAV-*Mir20b* led to elevated hepatic levels of *Mir20b* compared to AAV-Control injection in HFD-fed mice, and the level was slightly decreased by fenofibrate treatment (*Figure 7A*). Interestingly, we observed that administration of AAV-Control with fenofibrate increased the level of *Ppara*; however, fenofibrate could not restore the reduced *Ppara* expression by AAV-*Mir20b* (*Figure 7B*). Administration of fenofibrate reduced the body and liver weights of AAV-Control injected mice; however, AAV-*Mir20b* injected mice exhibited no significant differences by fenofibrate (*Figure 7C and D*). The ratio of fat to body weight also displayed no alterations between AAV-*Mir20b* and AAV-*Mir20b* with fenofibrate (*Figure 7E*). While the ratio of lean mass to body weight was increased by fenofibrate in AAV-*Mir20b* injected mice, the lean mass was comparable (*Figure 7F*). H&E staining, Oil Red O staining, NAS and hepatic TG levels demonstrated that fenofibrate significantly attenuated lipid accumulation in the liver of HFD-fed mice, but the effect of fenofibrate was suppressed by AAV-*Mir20b* (*Figure 7G–I*). Serum AST and ALT levels were decreased by fenofibrate, but this benefit was did not detected in AAV-*Mir20b* injected mice (*Figure 7J and K*). We further observed that blood glucose tolerance and insulin sensitivity were improved by fenofibrate; however, AAV-*Mir20b* offset the improvement by fenofibrate (*Figure 7L and M*). Fasting glucose, fasting insulin, and HOMA-IR levels were markedly decreased by fenofibrate in HFD-fed mice (*Figure 7N–P*). In AAV-*Mir20b* injected mice, fenofibrate did not reduce fasting insulin levels, but decreased fasting glucose and HOMA-IR levels. Fenofibrate also did not restore the suppressed expression of genes regulating FA β-oxidation in AAV-*Mir20b*-injected mice (*Figure 7Q*). Taken together, the effect of fenofibrate to ameliorate NAFLD-like symptoms was limited in AAV-*Mir20b* administrated HFD-fed mice because of the targeting PPARα by *Mir20b*.

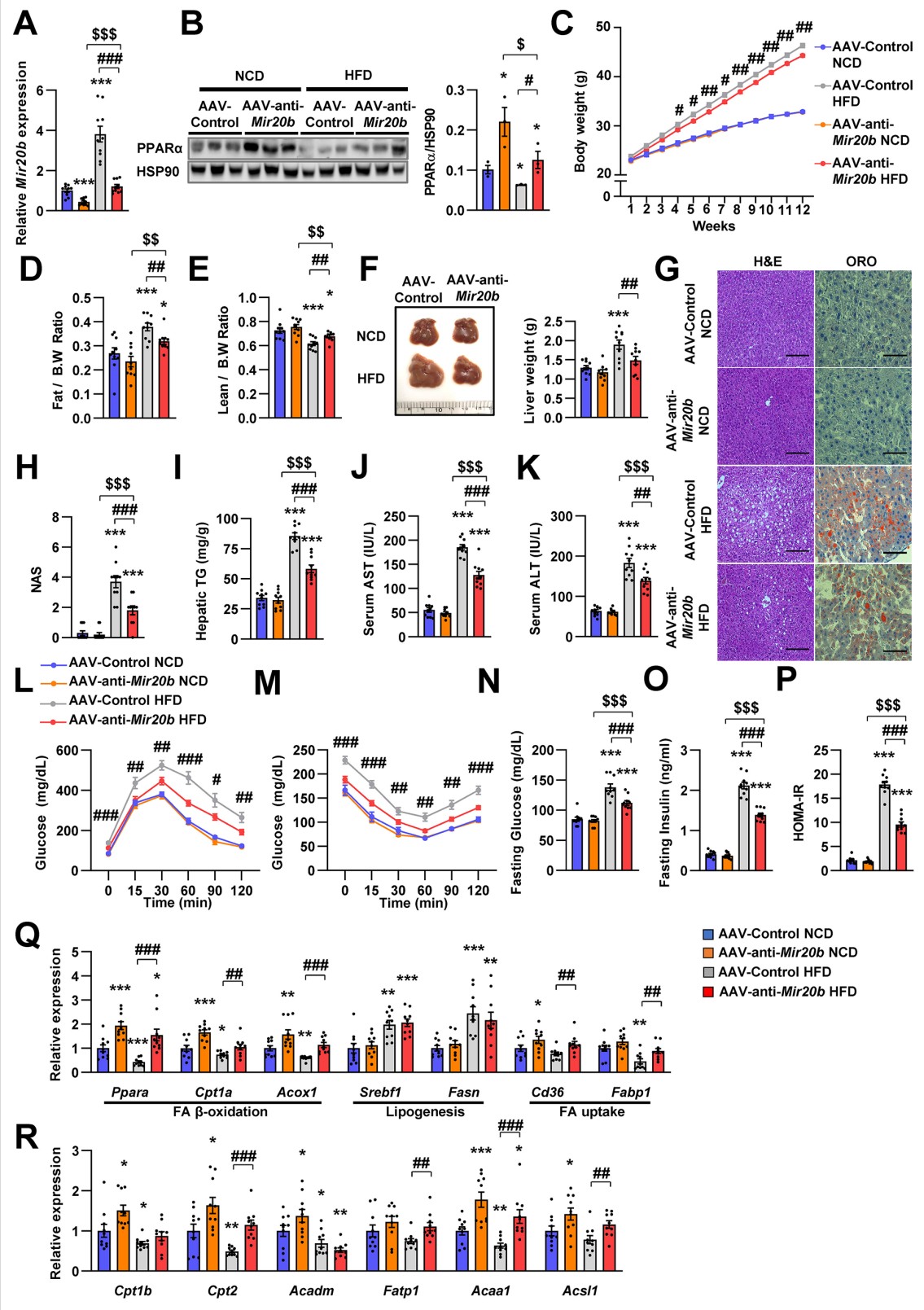

**Figure 6.** Inhibition of *Mir20b* alleviates hepatic steatosis in HFD-fed mice. C57BL/6 J mice were fed on normal chow diet (NCD, n = 10 per group) or high fat diet (HFD, n = 10 per group) for 12 weeks and administered with indicated AAVs. Hepatic expression of *Mir20b* (**A**) and PPARα (**B**) and body weight (**C**), the ratio of fat mass to body weight (**D**), the ratio of lean mass to body weight (**D**), representative images and weight of liver (**F**), representative images of H&E staining and Oil Red O staining of liver slides (**G**), NAFLD activity score (NAS) (**H**), hepatic TG (**I**), serum AST (**J**), and

*Figure 6 continued on next page*

Figure 6 continued

serum ALT (**K**), glucose tolerance (**L**), insulin tolerance (**M**), fasting glucose (**N**), fasting insulin (**O**), and HOMA-IR (**P**) were analyzed in indicated mice. Genes related to FA β-oxidation, lipogenesis and FA uptake (**Q**) and PPARα target genes (**R**) were determined by quantitative RT-PCR. Relative values are normalized to AAV-Control NCD. The intensity of PPARα blot was normalized to that of HSP90. Values represent means ± SEM (n = 10). *p < 0.05, **p < 0.01, ***p < 0.001 *vs* AAV-Control NCD. #p < 0.05, ##p < 0.01, ###p < 0.001 *vs* AAV-Control HFD. $p < 0.05, $$p < 0.01, $$$p < 0.001 *vs* AAV-anti-*Mir20b* NCD. Scale bar is 100 µm. Raw data can be found in a Source Data file named '*Figure 6—source data 1*'.

The online version of this article includes the following source data and figure supplement(s) for figure 6:

**Source data 1.** Inhibition of *Mir20b* alleviates hepatic steatosis in HFD-fed mice.

**Figure supplement 1.** The expression of *Mir20b* in adipose tissues of AAV injected mice.

**Figure supplement 1—source data 1.** The expression of *Mir20b* in adipose tissues of AAV injected mice.

**Figure supplement 2.** Inhibition of *Mir20b* significantly increases hepatic *Ppara* in HFD-fed mice.

**Figure supplement 2—source data 1.** Inhibition of *Mir20b* significantly increases hepatic *Ppara* in HFD-fed mice.

**Figure supplement 3.** The weight of adipose tissues of AAV injected mice.

**Figure supplement 3—source data 1.** The weight of adipose tissues of AAV injected mice.

**Figure supplement 4.** The expression of genes involved in lipolysis in adipose tissue and serum FFA in AAV injected mice.

**Figure supplement 4—source data 1.** The expression of genes involved in lipolysis in adipose tissue and serum FFA in AAV injected mice.

## *Mir20b* promotes liver inflammation and fibrosis in MCD-fed mice

We further demonstrated that RNA-seq data have a significant correlation of fold change values with previously published RNA-seq data under both NASH and liver fibrosis conditions within the NR transcription (*Hoang et al., 2019 Figure 8A and B*). This implies that *MIR20B* is able to set up an NR transcription program similar to that of NASH and liver fibrosis. To test this hypothesis, AAV-Control or AAV-anti-*Mir20b* was administered to C57BL/6 mice placed on a methionine/choline-deficient diet (MCD), which is the most widely used diet to induce NAFLD/NASH. Administration of AAV-anti-*Mir20b* led to decrease of *Mir20b* in the livers of NCD- and MCD-fed mice compared to AAV-Control injection (*Figure 8C*). We observed that the expression of *Ppara* was increased in MCD-fed mice and administration of AAV-anti-*Mir20b* displayed an elevation of PPARα, both at the mRNA and protein levels (*Figure 8D–F*). We next observed that AAV-anti-*Mir20b* administration significantly reduced hepatic steatosis in MCD-fed mice than in AAV-Control mice (*Figure 8G*). Liver sections clearly showed a decrease in both steatosis and fibrosis with AAV-anti-*Mir20b* administration in MCD-fed mice (*Figure 8H–J*). Consistently, AAV-anti-*Mir20b* administration decreased the levels of hepatic TG, AST, and ALT activity compared to AAV-Control injection (*Figure 8K–M*). Moreover, AAV-anti-*Mir20b* significantly reduced the expression of genes related to hepatic inflammation, including *Tnf* and *Il6* (*Figure 8N*), and fibrosis, including the NASH-relevant genes, such as *Acta2, Col1a1, Col3a1* and *Timp1* (*Figure 8O*), in MCD-fed mice. Taken together, these results indicate that *Mir20b* plays an important role in the development of fibrosis, inflammation, and hepatic steatosis in NAFLD progression.

## Anti-*Mir20b* enhances efficacy of fenofibrate in MCD-fed mice

Recently, drug development strategies for NAFLD/NASH are moving toward combination therapies (*Dufour et al., 2020*). However, the efficacy of developing drugs, including fenofibrate, against NAFLD/NASH is limited (*Fernández-Miranda et al., 2008*). Thus, we tested whether the combination of AAV-anti-*Mir20b* and fenofibrate would improve NAFLD in MCD-fed mice. The levels of hepatic *Mir20b* were reduced after administration of AAV-anti-*Mir20b* in MCD-fed mice compared to those in mice administered with AAV-Control, and this reduction was also observed after fenofibrate treatment (*Figure 9A*). Interestingly, the combination of AAV-anti-*Mir20b* and fenofibrate increased the levels of *Ppara* to a greater extent than AAV-anti-*Mir20b* alone (*Figure 9B and C*). AAV-anti-*Mir20b* or fenofibrate administration significantly reduced the liver weight and hepatic TG levels, and co-administration further reduced hepatic steatosis (*Figure 9D and E*). Histological sections showed that the combination of AAV-anti-*Mir20b* and fenofibrate improved NAFLD, as evidenced by the effects on both lipid accumulation and fibrosis in the liver (*Figure 9F–H*). Consistently, the levels of AST and ALT were significantly lower after combined treatment with AAV-anti-*Mir20b* and fenofibrate than after a single treatment (*Figure 9I and J*). In addition, the expression of genes related to hepatic

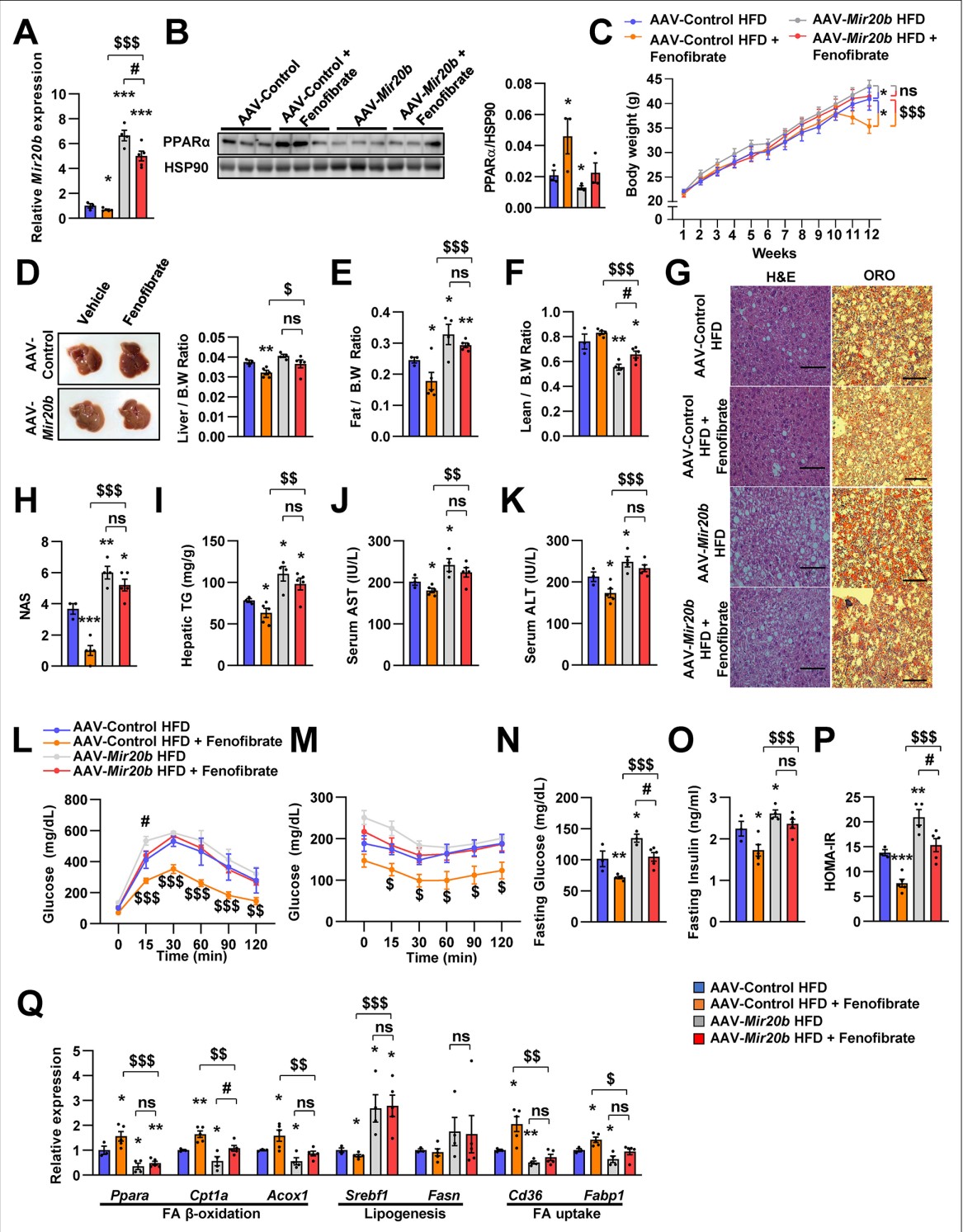

**Figure 7.** The effects of fenofibrate are limited in *Mir20b*-introduced mice. C57BL/6 J mice were fed on high fat diet (HFD) for 12 weeks and administered with indicated AAVs (n = 3–5 per group). Then, mice injected with vehicle or fenofibrate (100 mg/kg) for 4 weeks. Hepatic *Mir20b* (**A**) and PPARα expression (**B**), body weight (**C**), representative images and weight of liver (**D**), the ratio of fat mass to body weight (**E**), the ratio of lean mass to body weight (**F**), representative images of H&E staining and Oil Red O staining of liver slides (**G**), NAFLD activity score (NAS) (**H**), hepatic TG (**I**), serum AST (**J**), and serum ALT (**K**), glucose tolerance (**L**), insulin tolerance (**M**), fasting glucose (**N**), fasting insulin (**O**), and HOMA-IR (**P**) were analyzed in indicated mice. Genes related to FA β-oxidation, lipogenesis and FA uptake were determined by quantitative RT-PCR (**Q**). Relative values are normalized to AAV-Control HFD. The intensity of PPARα blot was normalized to that of HSP90. The intensity of PPARα blot was normalized to that of HSP90. Values represent means ± SEM (n = 3–5). ns, not significant, *p < 0.05, **p < 0.01, ***p < 0.001 *vs* AAV-Control HFD. #p < 0.05 *vs* AAV-*Mir20b* HFD. $p < 0.05,

*Figure 7 continued on next page*

*Figure 7 continued*

$^{\$\$}$p < 0.01, $^{\$\$\$}$p < 0.001 vs AAV-Control HFD+ fenofibrate. Scale bar is 100 μm. Raw data can be found in a Source Data file named '*Figure 7—source data 1*'.

The online version of this article includes the following source data for figure 7:

**Source data 1.** The effects of fenofibrate are limited in *Mir20b*-introduced mice.

---

inflammation, such as *Tnf* and *Il6* (*Figure 9K*), and fibrosis, such as *Acta2*, *Col1a1*, *Fn*, and *Timp1*, (*Figure 9L*), was further decreased by the combination of AAV-anti-*Mir20b* and fenofibrate. These results suggest that AAV-anti-*Mir20b* may increase the efficacy of fenofibrate, especially its effect on fibrosis, and provide a more effective option for improving NAFLD/NASH.

## Discussion

Obesity has been widely demonstrated to be central to the pathogenesis of NAFLD. Among other peripheral tissues, the liver plays a dominant role in the regulation of lipid homeostasis (*Pawlak et al., 2015*). Although abnormal regulation of metabolic homeostasis in the liver has been recognized in diabetes and NAFLD, the underlying molecular mechanisms remain to be elucidated. Growing evidence has demonstrated that *MIR20B* levels are significantly upregulated in the plasma miRNA profiles of NAFLD patients (*Jin et al., 2012*). Moreover, plasma *MIR20B* levels were highly elevated in T2DM/NAFLD patients compared to those in T2DM patients (*Ye et al., 2018*). However, the molecular mechanism through which *MIR20B* regulates NAFLD progression remains unknown. In this study, we demonstrated that *MIR20B* promotes NAFLD progression by modulating lipid metabolism, including FA β-oxidation, FA uptake, and TG synthesis, as well as ATP production by mitochondrial biogenesis. Our data clearly showed the regulatory mechanism of PPARα by *MIR20B*, and *MIR20B* may serve as a novel biological marker in NAFLD.

Decreased PPARα is contributed to development of NAFLD (*Francque et al., 2015*). A few miRNAs were reported to regulate *PPARA* expression in NAFLD. miR-34a targets *PPARA* and *SIRT1*, associating with FA oxidation and cholesterol synthesis. However, the effect of MIR34A on inflammation and fibrosis is not clear (*Ding et al., 2015*). MIR21 also decreases the expression of *PPARA* in NASH, however, activated PPARα by MIR21 suppression reduces inflammation, liver injury, and fibrosis without improvement in FA β-oxidation and lipid accumulation (*Loyer et al., 2016*). In the present study, we demonstrate the novel miRNA which has different mode of action. *MIR20B* showed the deteriorating effects on FA oxidation, steatosis, inflammation, and fibrosis in HFD- or MCD-fed mice. How these miRNAs targeting *PPARA* have different regulatory mechanisms should be further studied.

A previous study demonstrated that upregulated *MIR20B* levels in obesity-induced metabolic disorders such as T2DM were considered to prevent several targets, such as *STAT3*, *CD36*, and *PTEN*, which are involved in glucose and lipid homeostasis. The *MIR20B*/STAT axis, which is involved in the insulin signaling pathway, alters glycogen synthesis in human skeletal muscle cells (*Katayama et al., 2019*). Moreover, *MIR20B* directly targets *PTEN* involved in PI3K/Akt/mTOR signaling pathway that modulates glucose metabolism in gastric cells (*Streleckiene et al., 2020*). *Hosui et al., 2017* proposed a model in which the fatty acid transporter *CD36* is also a potential target of *MIR20B*, which crucially regulates hepatic lipid metabolism in *STAT5* KO mice models. But, it has been reported that *CD36* and *FABP1* are direct PPARα target genes (*Rakhshandehroo et al., 2010*). In this study, we observed that *MIR20B* downregulated *PPARA* and suppressed the expression of *CD36* and *FABP1*. Thus, *PPARA* is the primary target of *MIR20B* in regulating hepatic lipid metabolism.

*MIR20B* was induced by OA, but FA uptake was decreased by overexpression of *MIR20B*, and was accompanied by a considerable decrease in *CD36* expression (*Figure 4B and J*). However, other lipid transporters such as FATPs, except for PPARα target *FATP1*, were not significantly altered (*Figure 4— figure supplements 1* and *5*), suggesting that FA uptake is continued by these transporters. The expression of CD36 is relatively low in normal hepatocytes, and the molecule may not be the primary fatty acid transporter in these cells (*Wilson et al., 2016*). Furthermore, the decrease in FA uptake upon *CD36* KO is modest even during a HFD (*Wilson et al., 2016*). In addition, we observed that the expression of *MIR20B* is induced and increased for up to 24 hr by OA treatment (*Figure 4—figure supplement 6*). This is followed by a slight decrease, remaining at a constant elevated level. Together,

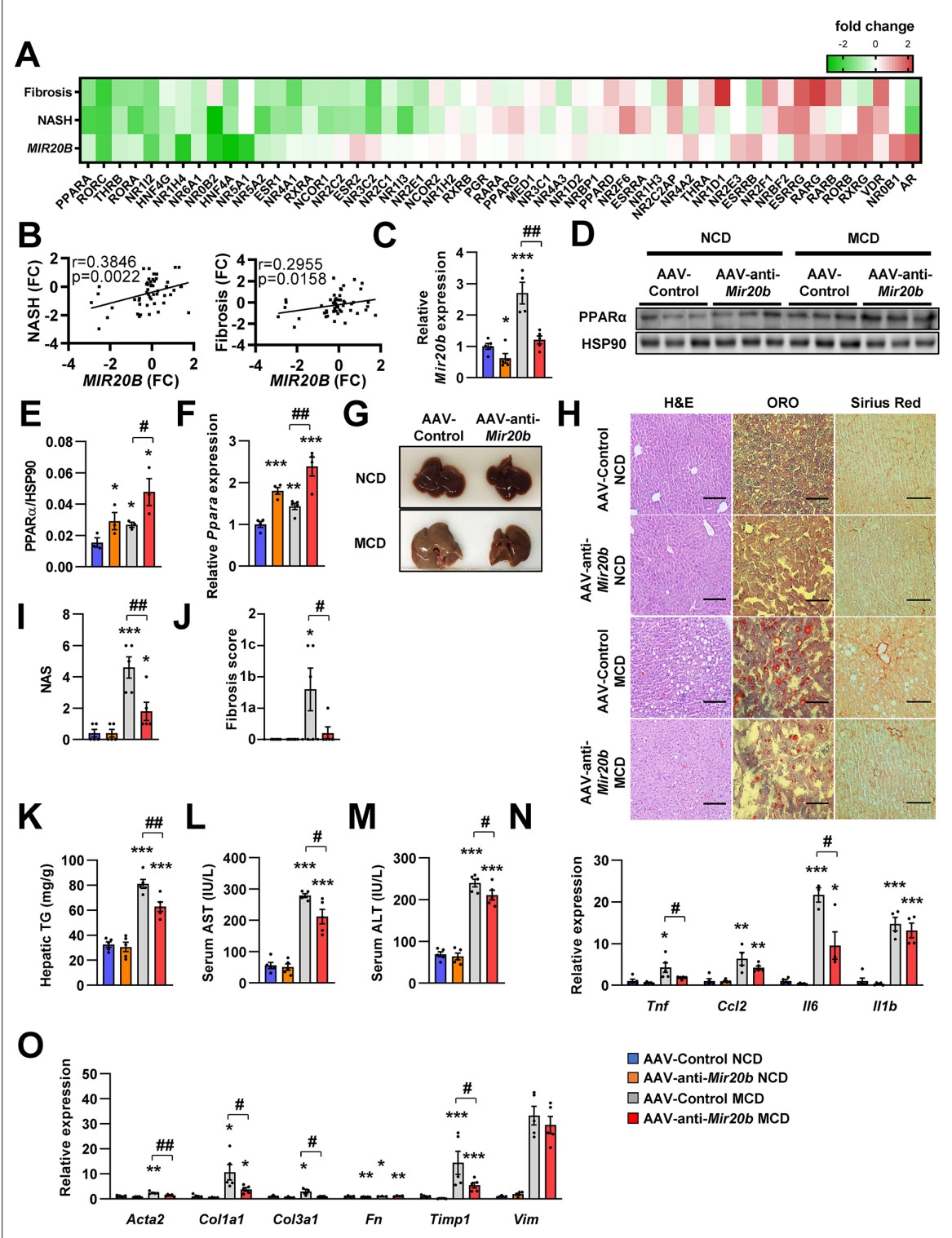

**Figure 8.** *Mir20b* promotes liver inflammation and fibrosis in MCD-fed mice. Heatmap (**A**) and correlation (**B**) of hepatic nuclear receptor gene expression in RNA-seq (*Figure 2*) with public databases of liver fibrosis or NASH patients. The values are fold change (FC) compared to each control samples. C57BL/6 J mice were fed on normal chow diet (NCD, n = 5 per group) or methionine-deficient diet (MCD, n = 5 per group) for 4 weeks and administered with indicated AAVs. Before 1 week of MCD challenge, mice were injected with AAV-Control or AAV-anti-*Mir20b*. Hepatic *Mir20b* (**C**) and PPARα expression (**D, E, F**), representative images of liver (**G**), H&E staining, Oil Red O, staining and Sirius Red staining of liver slides (**H**), NAFLD activity score (NAS) (**I**), fibrosis score (**J**), hepatic TG (**K**), serum AST (**L**), and serum ALT (**M**) were analyzed in indicated mice. Genes related to inflammation (**N**) and fibrosis (**O**) were determined by quantitative RT-PCR. Relative values are normalized to AAV-Control NCD. The intensity of PPARα blot was

*Figure 8 continued on next page*

*Figure 8 continued*

normalized to that of HSP90. Values represent means ± SEM (n = 5). *p < 0.05, **p < 0.01, ***p < 0.001 *vs* AAV-Control NCD. #p < 0.05, ##p < 0.01 *vs* AAV-Control MCD. Scale bar is 100 μm. Raw data can be found in a Source Data file named '*Figure 8—source data 1*'.

The online version of this article includes the following source data for figure 8:

**Source data 1.** *Mir20b* promotes liver inflammation and fibrosis in MCD-Fed Mice.

the findings indicated that other fatty acid transporters contributing to FA uptake account for the entry of OA into cells and *MIR20B* induction.

Hepatic steatosis could be affected by adipose tissue through FFA release and hepatic uptake of circulating FFAs (*Rasineni et al., 2021*). Our results showed that the epididymal adipose tissue of HFD-fed mice was enlarged upon AAV-*Mir20b* treatment; however, the serum FFA levels in these mice were comparable to those in mice treated with the AAV-Control (*Figure 5—figure supplement 4B*). Of note, the expression of genes related to lipolysis did not change in adipose tissues, and that of hepatic FA transporter, CD36, was decreased by AAV-*Mir20b* treatment (*Figure 5Q* and *Figure 5—figure supplement 4A*). In addition, excess hepatic TGs are secreted as very low-density lipoproteins (VLDLs), and the secretion rate increases with the TG level (*Fabbrini et al., 2008*). VLDLs deliver TGs from the liver to adipose tissue and contributes expansion of adipose tissue (*Chiba et al., 2003*). Together, these reports suggest that adipose tissue is also remodeled by the liver in HFD-fed mice and patients with NAFLD. Therefore, the increase of hepatic TGs by AAV-*Mir20b* is unlikely affected by AAV-*Mir20b* on epididymal adipose tissue, and the increase in fat content (*Figure 5—figure supplement 3*) may be a consequence of increased hepatic TG levels.

The expression of glycerolipid synthetic genes, including *AGPATs*, *GPATs*, and *DGATs*, was increased by OA treatment, but *MIR20B* overexpression did not influence the expression of these genes except for that of *DGAT1*. However, treatment with anti-*MIR20B* significantly reduced the expression of glycerolipid synthetic genes, including *GPATs* and *DGATs*, under OA treatment (*Figure 4C and N*). These results suggested that *MIR20B* is necessary but not sufficient to induce the expression of glycerolipid synthetic genes under OA treatment. We have shown that OA induces the expression of *MIR20B* (*Figure 1C and D*), which can explain why *MIR20B* overexpression did not show an additional enhancement under OA treatment. The increase in *DGAT1* expression induced by *MIR20B* might contribute to the increase in TG formation and capacity to store OA. This could change the flux of oleyl-CoA to TG synthesis, not β-oxidation with reduced expression of lipid oxidation-associated genes (*Figure 4B*). Thus, we can expect that the decrease in OA uptake and increase in TG formation induced by *MIR20B* resulted in reduced amounts of OA or oleyl-CoA inside the cell. However, as lipid consumption through FA oxidation is decreased by *MIR20B*, free OA or oleyl-CoA might be maintained at a stably increased level compared to that of OA-untreated MIR NC or *MIR20B* condition, and the impact of the changes in OA or oleyl-CoA levels on the transcriptional phenotype might not be significant as found in a constant elevated level of *MIR20B* by OA.

Recent reports suggest that some transcription factors regulate metabolic homeostasis by directly mediating the expression of miRNAs (*Yang and Wang, 2011*). E2F1, which is a member of the E2F transcription factor family, regulates myoblast differentiation and proliferation *via* the auto-regulatory feedback loop between E2F1 and *Mir20b* in muscle (*Luo et al., 2016*). Both the hepatic expression and activity of E2F1 are increased during obesity. *E2f1* deficiency protects against obesity- and diabetes-induced liver steatosis in mouse models (*Zhang et al., 2014*). Additionally, E2F1-induced chronic inflammation and hepatic lipid metabolism during NAFLD development (*Denechaud et al., 2016*). Consistent with these results, we observed that the expression of E2F1 was significantly increased in the fatty liver of both mice and humans, and its expression was positively correlated with that of *MIR20B* (*Figure 10*). These results suggest that E2F1 may be an upstream regulator of *MIR20B* in the liver, and this upregulation of *MIR20B* regulates lipid metabolism in the pathogenesis of NAFLD.

NAFLD patients can develop NASH, which is characterized by hepatic steatosis complicated by chronic hepatocellular damage and severe inflammation with fibrosis, potentially developing into cirrhosis and HCC (*Corey and Kaplan, 2014*). This indicates that suppression of NAFLD progression is the primary option for preventing the development of HCC. Recent reports suggest that *Ppara* KO mice fed an MCD developed much more severe NASH than wild-type mice, and the expression of *Ppara* in HCC tissue was significantly lower than that in normal liver tissue (*Montagner et al., 2016*). PPARα activation contributes to the inhibition of HCC cell proliferation Thus, hepatic PPARα plays a

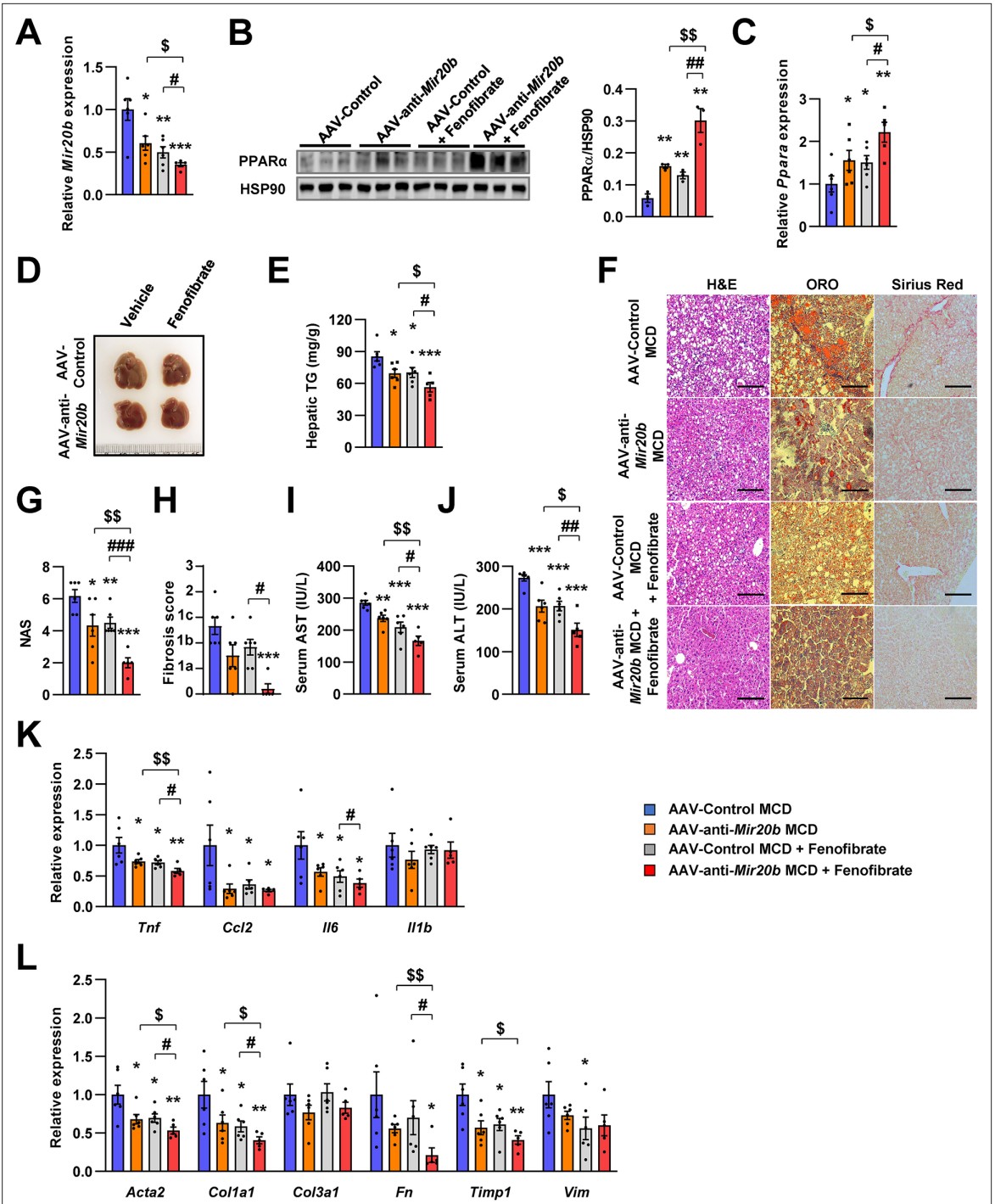

**Figure 9.** Inhibition of *Mir20b* with fenofibrate ameliorates liver inflammation and fibrosis in MCD-fed mice. C57BL/6 J were fed a methionine-deficient diet (MCD) for 4 weeks with administration of indicated AAVs (n = 5–6 per group). Then, mice injected with vehicle or fenofibrate (100 mg/kg) for 4 weeks. Hepatic *Mir20b* (**A**) and PPARA expression (**B, C**). Representative images of liver (**D**), hepatic TG (**E**), H&E staining, Oil Red O staining, Sirius Red staining of liver slides (**F**), NAFLD activity score (NAS) (**G**), fibrosis score (**H**), serum AST (**I**), and serum ALT (**J**) were analyzed in indicated mice. Genes related to inflammation (**K**) and fibrosis (**L**) were determined by quantitative RT-PCR. Relative values are normalized to AAV-Control MCD. The intensity of PPARα blot was normalized to that of HSP90. Values represent means ± SEM (n = 5–6). *p < 0.05, **p < 0.01, ***p < 0.001 *vs* AAV-Control MCD. #p < 0.05, ##p < 0.01, ###p < 0.001 *vs* AAV-Control MCD+ fenofibrate. $p < 0.05, $$p < 0.01 *vs* AAV-anti-*Mir20b* MCD. Scale bar is 100 µm. Raw data can be found in a Source Data file named '***Figure 9—source data 1***'.

The online version of this article includes the following source data for figure 9:

**Source data 1.** Inhibition of *Mir20b* with fenofibrate ameliorates liver inflammation and fibrosis in MCD-fed mice.

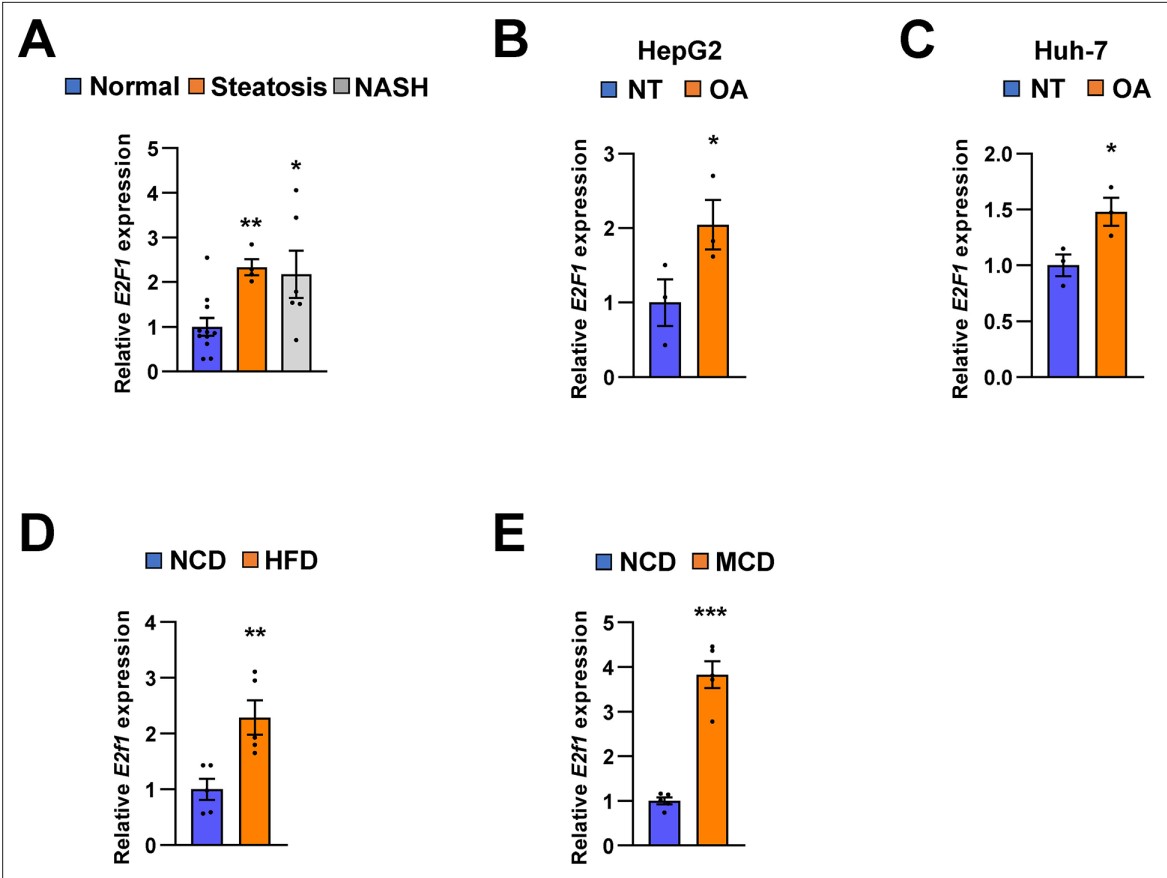

**Figure 10.** E2F1 is upregulated in both NAFLD patients and mice model. The expression of *E2F1* was analyzed by quantitative RT-PCR. Hepatic *E2F1* expression levels of steatosis or NASH patients were normalized to those of normal patients. *p < 0.05 and **p < 0.01 *vs* normal patients (**A**). *E2F1* expression levels from HepG2 cells (**B**) and Huh-7 cells (**C**) treated with OA for 24 hours were normalized to no treatment (NT). Hepatic *E2f1* expression levels from C57BL/6 J mice fed a HFD (**D**) and a MCD (**E**) were normalized to NCD. Values represent means ± SEM (n = 3–11). *p < 0.05, **p < 0.01, ***p < 0.001 *vs* NT in cells or NCD-fed mice, respectively. Raw data can be found in a Source Data file named '*Figure 10—source data 1*'.

The online version of this article includes the following source data for figure 10:

**Source data 1.** E2F1 is upregulated in both NAFLD patients and mice model.

crucial role in tumorigenesis in the liver (*Lefebvre et al., 2006*). Interestingly, it has been reported that upregulated *Mir20b* highly regulates cancer cell proliferation and promotes proliferation of H22 hepatocellular carcinoma cells (*Peng et al., 2019*; *Xia et al., 2020*). Thus, plasma *MIR20B* can be a promising target in liver cancer development. Indeed, we observed that the level of *MIR20B* was increased in NAFLD patients, but even robustly increased in the NASH stage. Furthermore, we observed that the hepatic function of *Mir20b* dramatically regulates the genes involved in inflammation and fibrosis by directly repressing *Ppara* in MCD-fed mice. Thus, our study strongly suggested that *MIR20B* regulates the pathogenesis of NAFLD, but might also be relevant in the development of severe stages of liver fibrosis and even in HCC.

Fibrates are FDA-approved medications for the treatment of patients with dyslipidemia (hypercholesterolemia, mixed dyslipidemia, hypertriglyceridemia) (*Katsiki et al., 2013*). Fibrates are well-known PPARα agonists, and PPARα is a widely accepted promising target for the treatment of NAFLD and NASH. In animal studies, fenofibrate improved hypertriglyceridemia and hepatic steatosis. In a small pilot trail, fenofibrate improved ALT and AST levels and hepatocellular ballooning degeneration, but did not show significant improvements in steatosis, lobular inflammation, fibrosis, or NAS score (*Fernández-Miranda et al., 2008*). Clofibrate, another PPARα agonist, did not show clinical benefit in the treatment of NASH (*Laurin et al., 1996*). Recently, the effect of omega-3 poly-unsaturated FAs (n-3 PUFA), which are ligands for PPARα, on patients with NAFLD was evaluated. Omacor, which consists of purified docosahexaenoic and eicosapentaenoic acids as ethyl esters, showed a linear decrease in

liver fat percentage without improvements in fibrosis score (*Scorletti et al., 2014*). Omacor has undergone a phase three clinical trial but the results have not been disclosed (NCT01277237). These results show that the effect of PPARα agonists on NAFLD is limited. Our results suggest one of the possible reasons for this limited effect. Here, we show that *PPARA* expression is suppressed by increased *MIR20B* expression, which may weaken the efficacy of fibrates. Indeed, our results showed that the effects of fenofibrate were limited in AAV-*Mir20b*-treated mice, and that the combination of fenofibrate and AAV-anti-*Mir20b* further improved the NAFLD phenotype in MCD-fed mice. Nowadays, it is generally accepted that treatment with PPARα agonists alone is limited and that a combination strategy is necessary for NAFLD. The inhibition of *MIR20B* presents an opportunity for the development of novel therapeutics for NAFLD.

## Acknowledgements

This research was funded by Korea Mouse Phenotyping Project (2016M3A9D5A01952411), the National Research Foundation of Korea (NRF) grant funded by the Korea government (2020R1F1A1061267, 2018R1A5A1024340, NRF-2021R1I1A2041463), the Future-leading Project Research Fund (1.210034.01) of UNIST to J.H.C., the National Research Foundation of Korea (NRF) grant funded by the Korea government (2020R1I1A1A01074940) to H-J.J., and the National Research Foundation of Korea (NRF) grant funded by the Korea government (2016M3C9A394589324) to D.N.

## Additional information

### Funding

| Funder | Grant reference number | Author |
| --- | --- | --- |
| National Research Foundation of Korea | 2020R1F1A1061267 | Jang Hyun Choi |
| National Research Foundation of Korea | 2018R1A5A1024340 | Jang Hyun Choi |
| National Research Foundation of Korea | NRF-2021R1I1A2041463 | Jang Hyun Choi |
| National Research Foundation of Korea | 2020R1I1A1A01074940 | Hyun-Jun Jang |
| Korea Mouse Phenotyping Project | 2016M3A9D5A01952411 | Jang Hyun Choi |
| Future-leading Project Research Fund | 1.210034.01 | Jang Hyun Choi |
| National Research Foundation of Korea | 2016M3C9A394589324 | Dougu Nam |

The funders had no role in study design, data collection and interpretation, or the decision to submit the work for publication.

### Author contributions

Yo Han Lee, Conceptualization, Data curation, Formal analysis, Investigation, Methodology, Project administration, Writing – original draft, Writing – review and editing; Hyun-Jun Jang, Conceptualization, Data curation, Formal analysis, Funding acquisition, Investigation, Methodology, Project administration, Writing – original draft, Writing – review and editing; Sounkou Kim, Data curation, Formal analysis, Methodology, Software, Visualization, Writing – original draft; Sun Sil Choi, Conceptualization, Data curation; Keon Woo Khim, Hye-jin Eom, Jimin Hyun, Kyeong Jin Shin, Investigation, Validation; Young Chan Chae, Hongtae Kim, Jiyoung Park, Supervision, Writing – review and editing; Neung Hwa Park, Resources, Validation, Writing – review and editing; Chang-Yun Woo, Chung Hwan Hong, Eun Hee Koh, Resources, Validation; Dougu Nam, Data curation, Software, Supervision, Validation, Writing – review and editing; Jang Hyun Choi, Conceptualization, Data curation, Funding acquisition, Project administration, Supervision, Writing – original draft, Writing – review and editing

## Author ORCIDs

Yo Han Lee http://orcid.org/0000-0002-3422-0306
Hyun-Jun Jang http://orcid.org/0000-0002-2261-0067
Jiyoung Park http://orcid.org/0000-0003-3705-4769
Dougu Nam http://orcid.org/0000-0003-0239-2899
Jang Hyun Choi http://orcid.org/0000-0003-0526-9028

## Ethics

Human subjects: Human liver tissue samples of 21 patients were acquired from the BioResource Center (BRC) of Asan Medical Center, Seoul, Republic of Korea. The process of 21 human tissue samples was officially approved by the Institutional Review Board of Asan Medical Center (IRB approval number: 2018-1512).

All animal experiments were performed according to procedures approved by the Ulsan National Institute of Science and Technology's Institutional Animal Care and Use Committee (UNISTIACUC-19-04).

## Decision letter and Author response

Decision letter https://doi.org/10.7554/eLife.70472.sa1
Author response https://doi.org/10.7554/eLife.70472.sa2

## Additional files

### Supplementary files

• Transparent reporting form

• Source data 1. Raw western blot data.

### Data availability

Sequencing data have been deposited in GEO under accession codes GSE168484. Other data generated or analysed during this study are included in the manuscript. Source data files have been provided.

The following dataset was generated:

| Author(s) | Year | Dataset title | Dataset URL | Database and Identifier |
|---|---|---|---|---|
| Kim S, Lee Y, Jang H | 2021 | Hepatic miR20b promotes nonalcholic fatty liver diseases by targeting PPARα | https://www.ncbi.nlm.nih.gov/geo/query/acc.cgi?acc=GSE168484 | NCBI Gene Expression Omnibus, GSE168484 |

The following previously published datasets were used:

| Author(s) | Year | Dataset title | Dataset URL | Database and Identifier |
|---|---|---|---|---|
| Hoang SA, Oseini A, Feaver RE, Cole B, Asgharpour A, Vincent R, Siddiqui M, Lawson MJ, Day NC, Taylor JM, Wamhoff BR, Mirshahi F, Contos MJ, Idowu M, Sanyal AJ | 2019 | Gene expression predicts histological severity and reveals distinct molecular profiles of nonalcoholic fatty liver disease | https://www.ncbi.nlm.nih.gov/geo/query/acc.cgi?acc=GSE130970 | NCBI Gene Expression Omnibus, GSE130970 |

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
