## [Editor Report]

The manuscript by Lee et al. provides mechanistic insight into the regulatory role of micro RNAs in modulating nuclear receptor expression and function. This is likely to have a high impact on the field as nuclear receptor regulation of metabolic disease is well established, however, the molecular mechanisms governing this process still remains unknown largely. Lee et al.'s manuscript provides a molecular target (miR-20b) that holds therapeutic potential in improving hepatic steatosis.

---

## [Decision Letter]

**Decision letter after peer review:**

Thank you for submitting your article "Hepatic miR-20b promotes nonalcoholic fatty liver disease by suppressing PPARα" for consideration by *eLife*. Your article has been reviewed by 3 peer reviewers, one of whom is a member of our Board of Reviewing Editors, and the evaluation has been overseen by Mone Zaidi as the Senior Editor. The following individual involved in review of your submission has agreed to reveal their identity: Suneil K Koliwad (Reviewer #2).

Essential revisions:

1) Add more replicates to your data to make sure that the effect on liver triglycerides are not part of a more global effect on organ and tissue weights in general or on body weight as a whole.

2) Share the populations characteristics from humans in figure 1 transparently so that we can know if the cohorts were very different in many potentially confounding ways or not.

*Reviewer #1 (Recommendations for the authors):*

To address the potential role of adipose off target expression of miR-20b in HFD mice in contributing to hepatic steatosis experimentally, the authors should assess circulating free fatty acids in miR-20b treated mice to determine if enhanced lipolysis in adipose tissue contributes to hepatic steatosis.

Assessing fatty acid oxidation or oil red O staining in mutated 3'UTR PPARa would show that miR-20b regulation of PPARa 3'UTR is the mechanism regulating hepatic lipid metabolism.

*Reviewer #2 (Recommendations for the authors):*

1. Try to validate your data more fully. A. include other miRs and other NRs to compare vs. miR-20b and PPARalpha, respectively. Are the effects of your stated targets specific, or do other miRs and NRs from your screen also have impact on NAFLD biology? Similarly, try to validate your cell-based data on a wider array of genes in vivo, best with bulk RNAseq or a PPARalpha target gene profiling.

2. Add more replicates to your data to make sure that the effect on liver triglycerides are not part of a more global effect on organ and tissue weights in general or on body weight as a whole.

3. Match the groups better, including human cohorts as well as the people in the two data sets. Or at least share the populations characteristics from humans in figure 1 transparently so that we can know if the cohorts were very different in many potentially confounding ways or not.

*Reviewer #3 (Recommendations for the authors):*

1. In Figure 1, more information on the human liver tissue samples would be helpful. From the methods section, it seems at least some of these samples were adjacent to tumor, but it is not entirely clear this is the case. If some or all are adjacent to tumor, might the presence of tumor affect the expression of miR-20b? Which samples are adjacent to tumor? What kinds of tumors are there in these livers? How was the tissue selected and dissected-was this done by a pathologist? How were the diagnoses of control, steatosis, and NASH arrived at? It is odd that only 1/4 patients in the NASH group is overweight and none are obese.

2. More information on the mouse models in Figure 1 would be helpful as well. Are the mice overweight/obese at the time the samples were taken? Do their livers show steatosis, fibrosis, and/or inflammation?

3. I think the a critical experiment is missing-what happens when anti-mIR-20b is combined with fenofibrate in MCD-fed mice?

---

## [Author Response]

Essential revisions:1) Add more replicates to your data to make sure that the effect on liver triglycerides are not part of a more global effect on organ and tissue weights in general or on body weight as a whole.

Thank you for your valuable comment. To clarify the effect of *MIR20B*, we increased the number of samples in revised Figure 1B, Figure 5, and Figure 6. In addition, we advanced the timing of adeno-associated virus (AAV)-*Mir20b* or AAV-anti-*Mir20b* administration to mice. Previously, we introduced AAV treatment after four weeks of high fat diet (HFD) feeding. Ectopic *Mir20b* expression or anti-*Mir20b* treatment significantly changed the expression of *PPARA*; however, the effect on the pathophysiological properties of the liver was significant but modest. We thought that this was because there was not enough time to make a proper impact on the liver. Thus, to maximize the effect of *Mib20b*, the AAV was administered when the HFD was started. The new results showed more significant effects of *Mib20b*.

As per the concerns of the reviewers, hepatic steatosis could be affected by adipose tissue through free fatty acid (FFA) release and hepatic uptake of circulating FFAs (Rasineni et al., 2021). Our results showed that the epididymal adipose tissue of HFD-fed mice was enlarged upon AAV-*Mir20b* treatment; however, the serum FFA levels in these mice were comparable to those in mice treated with the AAV-Control. Of note, the expression of genes related to lipolysis did not change in adipose tissues, and that of hepatic FA transporter, *CD36*, was decreased by AAV-*Mir20b* treatment (Figure 5Q and Figure 5—figure supplement 4). In addition, excess hepatic triglycerides (TGs) are secreted as very low density lipoproteins (VLDLs), and the secretion rate increases with the TG level (Fabbrini et al., 2008). VLDLs deliver TGs from the liver to adipose tissue and contributes expansion of adipose tissue (Chiba, Nakazawa, Yui, Kaneko, and Shimokado, 2003). Together, these reports suggest that adipose tissue is also remodeled by the liver in HFD-fed mice and non-alcoholic fatty liver disease (NAFLD) patients. Therefore, the levels of hepatic TGs are unlikely affected by epididymal adipose tissue, and the increase in fat content may be a consequence of increased hepatic TG levels. We have added these points in Discussion (page 25-26, line 596-608).

2) Share the populations characteristics from humans in figure 1 transparently so that we can know if the cohorts were very different in many potentially confounding ways or not.

Thank you for your comment. Accordingly, we have included the patient information transparently, including fasting glucose and total cholesterol in a table (Figure 1—figure supplement 1A, B). To increase the statistical power and prevent confounding effects, we increased the number of samples and tried to match them to compare age, weight, and male/female ratio between the groups. Due to the limited number of patient samples, the cohorts could not be perfectly matched. Nevertheless, there were no significant differences in age and male/female ratio among the three groups. Specifically, serum AST, ALT, and fasting glucose levels were significantly increased with progression from normal to non-alcoholic steatohepatitis (NASH), but total cholesterol was comparable as previously reported (Chung et al., 2020).

Reviewer #1 (Recommendations for the authors):To address the potential role of adipose off target expression of miR-20b in HFD mice in contributing to hepatic steatosis experimentally, the authors should assess circulating free fatty acids in miR-20b treated mice to determine if enhanced lipolysis in adipose tissue contributes to hepatic steatosis.

We observed that slight increase of *Mir20b* expression in epididymal adipose tissue of AAV-*miR20b* HFD-fed mice compared to AAV-control NCD-fed mice, not HFD-fed mice. The expression of *Mir20b* in adipose tissue of between AAV-control HFD and AAV-*Mir20b* HFD mice was not significantly changed (Figure 5—figure supplement1).

We have revised the text and added the discussion about the potential role of adipose tissue (page 26, line 596-608). Hepatic steatosis could be affected by adipose tissue through FFA release and hepatic uptake of circulating FFAs (Rasineni et al., 2021). Our results showed that the epididymal adipose tissue of HFD-fed mice was enlarged upon AAV-*Mir20b* treatment; however, the serum FFA levels in these mice were comparable to those in mice treated with the AAV-Control. Of note, the expression of genes related to lipolysis did not change in adipose tissues, and that of hepatic FA transporter, *CD36*, was decreased by AAV-*Mir20b* treatment (Figure 5Q and Figure 5—figure supplement 4). In addition, excess hepatic TGs are secreted as VLDLs, and the secretion rate increases with the TG level (Fabbrini et al., 2008). VLDLs deliver TGs from the liver to adipose tissue and contributes expansion of adipose tissue (Chiba et al., 2003). Together, these reports suggest that adipose tissue is also remodeled by the liver in HFD-fed mice and non-alcoholic fatty liver disease (NAFLD) patients. Therefore, the levels of hepatic TGs are unlikely affected by epididymal adipose tissue, and the increase in fat content may be an incidental consequence of increased hepatic TG levels.

Assessing fatty acid oxidation or oil red O staining in mutated 3'UTR PPARa would show that miR-20b regulation of PPARa 3'UTR is the mechanism regulating hepatic lipid metabolism.

To provide functional evidence, we tried to establish the *PPARA* 3’UTR mutation knock-in (KI) system in cells. However, we could not succeed because of technical difficulties and time constraints. Alternatively, we introduced the wild type *PPARA* open reading frame (ORF) followed by either the wild type (WT) or mutant (Mut) 3’UTR of *PPARA* in HepG2 cells, and analyzed the importance of the 3’UTR of *PPARA*. As shown in Figure 2—figure supplement 5C, *MIR20B* significantly suppressed the expression of *PPARA* and its target genes in *PPARA*-3’UTR WT expressing cells. Furthermore, Oil Red O staining showed that *MIR20B* expression increased the intracellular lipid content in these cells (Figure 2—figure supplement 5B). However, *MIR20B* did not have an effect on either the expression of *PPARA* and its target genes or intracellular lipid content in *PPARA*-3’UTR Mut expressing cells (Figure 2—figure supplement 5C, D). We have added the new results in page 18, line 359-366 and Figure 2—figure supplement 5.

Reviewer #2 (Recommendations for the authors):1. Try to validate your data more fully. A. include other miRs and other NRs to compare vs. miR-20b and PPARalpha, respectively. Are the effects of your stated targets specific, or do other miRs and NRs from your screen also have impact on NAFLD biology? Similarly, try to validate your cell-based data on a wider array of genes in vivo, best with bulk RNAseq or a PPARalpha target gene profiling.

Thank you for your comment.

(A) This is a very good point. In the analysis of the regulatory network, other miRNAs including MIR129 and MIR106A appeared to possibly regulate nuclear receptors in NAFLD. We further confirmed the relationship between candidate miRNAs and NAFLD progression in patient samples. As shown in the revised Figure 1B, we observed that the expression of *MIR20B* was more robustly and significantly changed with NAFLD progression than that of MIR129 and MIR106A. This tendency was also confirmed in other experiments using OA-treated HepG2 and Huh7 cells or HFD-fed mice (Figure 1—figure supplement 4). Thus, we focused on the role of *MIR20B* in NAFLD. Nevertheless, we do not rule out the possibility that other miRNAs may be involved in NAFLD progression. Subsequent studies may uncover the roles of other miRNAs in liver physiology.

(B) We also observed that other nuclear receptors, such as *RORA*, *RORC*, and *THRB*, could be potential targets of *MIR20B* (Figure 2H and Figure 2—figure supplement 3). However, in the patient data, there was no significant correlation between the expression of those nuclear receptors and that of *MIR20B*. In addition, among the candidate targets, only *PPARA* was selected as an overlapped predicted target of *MIR20B* by various miRNA target prediction programs, including miRDB, picTAR, TargetSCAN, and miRmap (Figure 2J, Figure 2—figure supplement 2). Consistent with these results, we observed that *PPARA*, not other nuclear receptors is the target gene of *Mib20b* in both AAV-*Mir20b* and AAV-anti-*Mir20b* mice (Figure 5—figure supplement 2, Figure 6—figure supplement 2). Thus, we focused on *PPARA* as a *MIR20B* target in NAFLD.

(C) Accordingly, we selected PPARα target genes altered by *MIR20B* in OA-treated cells (Figure 4—figure supplement 1A, B), and then examined the hepatic expression of PPARα target genes in HFD-fed mice treated with AAV-*Mir20b* or AAV-anti-*Mir20b* (Figure 5R and 6R). The expression of most PPARα target genes was decreased by OA treatment and the HFD, and AAV-*Mir20b* treatment further reduced their expression. In contrast, AAV-anti-*Mir20b* treatment rescued the reduced expression of PPARα target genes under OA treatment and the HFD. These results suggested that *Mib20b* suppresses PPARα in vivo, which is consistent with the results from cells. We have added these results in Figure 4—figure supplement 1A, B, Figure 5R, and Figure 6R.

2. Add more replicates to your data to make sure that the effect on liver triglycerides are not part of a more global effect on organ and tissue weights in general or on body weight as a whole.

Thank you for your comment.

Accordingly, we conducted additional in vivo experiments with larger n values (n = 10). Then, we replaced the liver images with more representative ones. AAV-*Mir20b* robustly induced the hepatic expression of *Mir20b* and significantly increased the liver weight and hepatic TG levels (Figure 5F, I). In the liver of normal human, intrahepatic TGs do not exceed 5 % of the liver weight (Fabbrini and Magkos, 2015). In our results, TG levels were increased more than three times by the HFD, but the impact on liver weight was limited, as TGs did not account for more than 10 % of the liver weight (Figure 5I).

Excess hepatic TGs are secreted as very low density lipoproteins (VLDLs), and the secretion rate increases with the TG level (Fabbrini et al., 2008). VLDLs deliver TGs from the liver to adipose tissue and other metabolic tissues (Heeren and Scheja, 2021). The excess hepatic TGs induced by AAV-*Mir20b* were presumably transferred to epididymal adipose tissue, contributing to the increase in adipose tissue weight, while inguinal and brown adipose tissues were not significantly affected by AAV-*Mir20b* (Figure 5—figure supplement 3). Together, the fat mass measured by EchoMRI included intrahepatic and adipose TGs, and mirrored the increases shown in Figure 5D. In addition, AAV-*Mir20b* induced the expression of hepatic *DGAT1*, which could explain increased TG secretion through VLDLs (Figure 4C) (Alves-Bezerra and Cohen, 2017; Liang et al., 2004).

Conversely, the supply of FFAs from adipose tissue might have contributed to hepatic steatosis. However, we observed that there were no significant changes in the expression of *Mir20b* and lipolytic genes in adipose tissue (Figure 5—figure supplement 4A). Furthermore, the serum FFA levels in the AAV-Control and AAV-*Mir20b* groups under the HFD were comparable (Figure 5—figure supplement 4B). These findings suggested that increased intrahepatic TG levels constituted the specific and primary effect of AAV-*Mir20b*.

3. Match the groups better, including human cohorts as well as the people in the two data sets. Or at least share the populations characteristics from humans in figure 1 transparently so that we can know if the cohorts were very different in many potentially confounding ways or not.

In Figure 1, we analyzed the differential expression of NR in NAFLD using public GSE data (GSE130970) consisting of patients with NAFLD and age- and weight-matched normal controls (Hoang et al., 2019). To verify the expression of *MIR20B,* we assessed the miRNA levels in patients and normal controls from the Asan Medical Center (Seoul, Republic of Korea), who were diagnosed by pathologists and age- and weight-matched. The population characteristics of GSE130970 are publicly available on the GEO website and the characteristics of the patients from the Asan Medical Center have been provided in Figure 1—figure supplement 1.

Reviewer #3 (Recommendations for the authors):1. In Figure 1, more information on the human liver tissue samples would be helpful. From the methods section, it seems at least some of these samples were adjacent to tumor, but it is not entirely clear this is the case. If some or all are adjacent to tumor, might the presence of tumor affect the expression of miR-20b? Which samples are adjacent to tumor? What kinds of tumors are there in these livers? How was the tissue selected and dissected-was this done by a pathologist? How were the diagnoses of control, steatosis, and NASH arrived at? It is odd that only 1/4 patients in the NASH group is overweight and none are obese.

To verify the expression of *Mir20b*, we assessed the miRNA levels in steatosis, NASH and controls from the Asan Medical Center (Seoul, Republic of Korea), which were diagnosed by pathologists and age- and weight-matched. Population characteristics are provided in Figure 1—figure supplement 1. Human samples used in this study were from hepatocellular carcinomas (HCC) liver resection. All samples were taken from non-tumor parts of the resected tissue and selected through histological analysis of pathologists using the grading and staging system for NASH (Brunt, Janney, Di Bisceglie, Neuschwander-Tetri, and Bacon, 1999). Although limitations in completely excluding the effect of HCC, histologically normal tissue adjacent to the tumor has obvious distinctions in global transcriptome patterns for the tumor and can be used as a control (Aran et al., 2017). We have revised the text (page 8, line 121-133).

Although it is well known that NAFLD is closely related to obesity, type 2 diabetes, and metabolic syndrome, some non-obese cases can also cause NAFLD. Non-obese NAFLD is typically associated with a BMI < 25 kg / m2 (in Asians < 23 kg / m2), approximately 5 – 26 % of the Asian population and 7 – 20 % of the Western population (Younes and Bugianesi, 2019). The pathophysiological mechanism of NAFLD of the "lean" phenotype has not been fully elucidated. However, it was more closely related to visceral fat than to subcutaneous fat and affected by body composition changes such as decreased muscle mass, genetic predisposition, and changes in gut microbiota patterns (Kuchay, Martinez-Montoro, Choudhary, Fernandez-Garcia, and Ramos-Molina, 2021). Visceral fat is associated with insulin resistance, hypertriglyceridemia, and low HDL cholesterol, which increases pro-inflammatory cytokine levels and circulating free fatty acid levels, resulting in hepatic fat accumulation and subsequent liver damage (Mirza, 2011; van der Poorten et al., 2008). These may explain the pathogenesis of metabolic syndrome and NAFLD in non-obese individuals with relatively low BMI (Liu, 2012).

2. More information on the mouse models in Figure 1 would be helpful as well. Are the mice overweight/obese at the time the samples were taken? Do their livers show steatosis, fibrosis, and/or inflammation?

Thank you for your comment. Accordingly, gene expression related to steatosis, fibrosis, and/or inflammation has been added in the revised Figure 1—figure supplement 3. HFD-fed mice and *Lep^ob/ob^* mice were obese, but MCD-fed mice lost weight compared with NCD-fed mice. HFD-fed, MCD-fed, and *Lep^ob/ob^* mice showed significant increased steatosis and inflammation, and MCD-fed mice clearly showed fibrosis in the liver.

3. I think the a critical experiment is missing-what happens when anti-mIR-20b is combined with fenofibrate in MCD-fed mice?

We tested whether the combination of anti-*Mir20b* and fenofibrate would improve NAFLD in MCD-fed mice. The levels of hepatic *Mir20b* were reduced after administration of AAV-anti-*Mir20b* in MCD-fed mice compared to those in mice administered with AAV-Control, and this reduction was also observed after fenofibrate treatment (Figure 9A). Interestingly, the combination of AAV-anti-*Mir20b* and fenofibrate increased the levels of PPARα to a greater extent than AAV-*Mir20b* alone (Figure 9B, C). AAV-anti-*Mir20b* or fenofibrate administration significantly reduced the liver weight and hepatic TG levels, and co-administration further reduced hepatic steatosis (Figure 9D, E). Histological sections showed that the combination of AAV-anti-*Mir20b* and fenofibrate improved NAFLD, as evidenced by the effects on both lipid accumulation and fibrosis in the liver (Figure 9F-H). Consistently, the levels of AST and ALT were significantly lower after combined treatment with AAV-anti-*Mir20b* and fenofibrate than after a single treatment (Figure 9I, J). In addition, the expression of genes related to hepatic inflammation, such as *Tnf* and *Il6* (Figure 9K), and fibrosis, such as *Acta2*, *Col1a1*, *Fn*, and *Timp1*, (Figure 9L), was further decreased by the combination of AAV-anti-*Mir20b* and fenofibrate. These results suggest that AAV-anti-*Mir20b* may increase the efficacy of fenofibrate, especially its effect on fibrosis, and provide a more effective option for improving NAFLD/NASH.

**References**

Alves-Bezerra, M., and Cohen, D. E. (2017). Triglyceride Metabolism in the Liver. *Compr Physiol, 8*(1), 1-8. doi:10.1002/cphy.c170012

Aran, D., Camarda, R., Odegaard, J., Paik, H., Oskotsky, B., Krings, G.,... Butte, A. J. (2017). Comprehensive analysis of normal adjacent to tumor transcriptomes. *Nat Commun, 8*(1), 1077. doi:10.1038/s41467-017-01027-z

Brunt, E. M., Janney, C. G., Di Bisceglie, A. M., Neuschwander-Tetri, B. A., and Bacon, B. R. (1999). Nonalcoholic steatohepatitis: a proposal for grading and staging the histological lesions. *Am J Gastroenterol, 94*(9), 2467-2474. doi:10.1111/j.1572-0241.1999.01377.x

Chiba, T., Nakazawa, T., Yui, K., Kaneko, E., and Shimokado, K. (2003). VLDL induces adipocyte differentiation in ApoE-dependent manner. *Arterioscler Thromb Vasc Biol, 23*(8), 1423-1429. doi:10.1161/01.ATV.0000085040.58340.36

Chung, G. E., Yim, J. Y., Kim, D., Kwak, M. S., Yang, J. I., Park, B.,... Kim, J. S. (2020). Nonalcoholic Fatty Liver Disease Is Associated with Benign Prostate Hyperplasia. *J Korean Med Sci, 35*(22), e164. doi:10.3346/jkms.2020.35.e164

Cui, S., Pan, X. J., Ge, C. L., Guo, Y. T., Zhang, P. F., Yan, T. T.,... Wang, H. (2021). Silybin alleviates hepatic lipid accumulation in methionine-choline deficient diet-induced nonalcoholic fatty liver disease in mice via peroxisome proliferator-activated receptor α. *Chin J Nat Med, 19*(6), 401-411. doi:10.1016/S1875-5364(21)60039-0

Dufour, J. F., Caussy, C., and Loomba, R. (2020). Combination therapy for non-alcoholic steatohepatitis: rationale, opportunities and challenges. *Gut, 69*(10), 1877-1884. doi:10.1136/gutjnl-2019-319104

Fabbrini, E., and Magkos, F. (2015). Hepatic Steatosis as a Marker of Metabolic Dysfunction. *Nutrients, 7*(6), 4995-5019. doi:10.3390/nu7064995

Fabbrini, E., Mohammed, B. S., Magkos, F., Korenblat, K. M., Patterson, B. W., and Klein, S. (2008). Alterations in adipose tissue and hepatic lipid kinetics in obese men and women with nonalcoholic fatty liver disease. *Gastroenterology, 134*(2), 424-431. doi:10.1053/j.gastro.2007.11.038

Fernandez-Miranda, C., Perez-Carreras, M., Colina, F., Lopez-Alonso, G., Vargas, C., and Solis-Herruzo, J. A. (2008). A pilot trial of fenofibrate for the treatment of non-alcoholic fatty liver disease. *Dig Liver Dis, 40*(3), 200-205. doi:10.1016/j.dld.2007.10.002

Heeren, J., and Scheja, L. (2021). Metabolic-associated fatty liver disease and lipoprotein metabolism. *Mol Metab, 50*, 101238. doi:10.1016/j.molmet.2021.101238

Hoang, S. A., Oseini, A., Feaver, R. E., Cole, B. K., Asgharpour, A., Vincent, R.,... Sanyal, A. J. (2019). Gene Expression Predicts Histological Severity and Reveals Distinct Molecular Profiles of Nonalcoholic Fatty Liver Disease. *Sci Rep, 9*(1), 12541. doi:10.1038/s41598-019-48746-5

Katsiki, N., Nikolic, D., Montalto, G., Banach, M., Mikhailidis, D. P., and Rizzo, M. (2013). The role of fibrate treatment in dyslipidemia: an overview. *Curr Pharm Des, 19*(17), 3124-3131. doi:10.2174/1381612811319170020

Kuchay, M. S., Martinez-Montoro, J. I., Choudhary, N. S., Fernandez-Garcia, J. C., and Ramos-Molina, B. (2021). Non-Alcoholic Fatty Liver Disease in Lean and Non-Obese Individuals: Current and Future Challenges. *Biomedicines, 9*(10). doi:10.3390/biomedicines9101346

Laurin, J., Lindor, K. D., Crippin, J. S., Gossard, A., Gores, G. J., Ludwig, J.,... McGill, D. B. (1996). Ursodeoxycholic acid or clofibrate in the treatment of non-alcohol-induced steatohepatitis: a pilot study. *Hepatology, 23*(6), 1464-1467. doi:10.1002/*hep*.510230624

Liang, J. J., Oelkers, P., Guo, C., Chu, P. C., Dixon, J. L., Ginsberg, H. N., and Sturley, S. L. (2004). Overexpression of human diacylglycerol acyltransferase 1, acyl-coa:cholesterol acyltransferase 1, or acyl-CoA:cholesterol acyltransferase 2 stimulates secretion of apolipoprotein B-containing lipoproteins in McA-RH7777 cells. *J Biol Chem, 279*(43), 44938-44944. doi:10.1074/jbc.M408507200

Liu, C. J. (2012). Prevalence and risk factors for non-alcoholic fatty liver disease in Asian people who are not obese. *J Gastroenterol Hepatol, 27*(10), 1555-1560. doi:10.1111/j.1440-1746.2012.07222.x

Mirza, M. S. (2011). Obesity, Visceral Fat, and NAFLD: Querying the Role of Adipokines in the Progression of Nonalcoholic Fatty Liver Disease. *ISRN Gastroenterol, 2011*, 592404. doi:10.5402/2011/592404

Rakhshandehroo, M., Hooiveld, G., Muller, M., and Kersten, S. (2009). Comparative analysis of gene regulation by the transcription factor PPARalpha between mouse and human. *PLoS One, 4*(8), e6796. doi:10.1371/journal.pone.0006796

Rakhshandehroo, M., Knoch, B., Muller, M., and Kersten, S. (2010). Peroxisome proliferator-activated receptor α target genes. *PPAR Res, 2010*. doi:10.1155/2010/612089

Rasineni, K., Jordan, C. W., Thomes, P. G., Kubik, J. L., Staab, E. M., Sweeney, S. A.,... Casey, C. A. (2021). Contrasting Effects of Fasting on Liver-Adipose Axis in Alcohol-Associated and Non-alcoholic Fatty Liver. *Front Physiol, 12*, 625352. doi:10.3389/fphys.2021.625352

Scorletti, E., Bhatia, L., McCormick, K. G., Clough, G. F., Nash, K., Hodson, L.,... Study, W. (2014). Effects of purified eicosapentaenoic and docosahexaenoic acids in nonalcoholic fatty liver disease: results from the Welcome* study. *Hepatology, 60*(4), 1211-1221. doi:10.1002/*hep*.27289

van der Poorten, D., Milner, K. L., Hui, J., Hodge, A., Trenell, M. I., Kench, J. G.,... George, J. (2008). Visceral fat: a key mediator of steatohepatitis in metabolic liver disease. *Hepatology, 48*(2), 449-457. doi:10.1002/*hep*.22350

Wilson, C. G., Tran, J. L., Erion, D. M., Vera, N. B., Febbraio, M., and Weiss, E. J. (2016). Hepatocyte-Specific Disruption of CD36 Attenuates Fatty Liver and Improves Insulin Sensitivity in HFD-Fed Mice. *Endocrinology, 157*(2), 570-585. doi:10.1210/en.2015-1866

Younes, R., and Bugianesi, E. (2019). NASH in Lean Individuals. *Semin Liver Dis, 39*(1), 86-95. doi:10.1055/s-0038-1677517